# CoRA: Covariate-Aware Adaptation of Time Series Foundation Models

## Abstract

Time Series Foundation Models (TSFMs) have shown significant impact through their model capacity, scalability, and zero-shot generalization. However, due to the heterogeneity of inter-variate dependencies and the backbone scalability on large-scale multivariate datasets, most TSFMs are typically pre-trained on univariate time series. This limitation renders them oblivious to crucial information from diverse covariates in real-world forecasting tasks. To further enhance the performance of TSFMs, we propose a general **Co**variate-awa**R**e **A**daptation (**CoRA**) framework for TSFMs. It leverages pre-trained backbones of foundation models while effectively incorporating exogenous covariates from various modalities, including time series, language, and images, to improve the quality of predictions. Technically, CoRA maintains the equivalence of initialization and parameter consistency during adaptation. With preserved backbones of foundation models as frozen feature extractors, the outcome embeddings from foundation models are empirically demonstrated more informative than raw data. Further, CoRA employs a novel Causality Embedding to automatically evaluate covariates regarding their causal predictability with respect to the target variate. We incorporate these weighted embeddings with a zero-initialized condition-injection mechanism, avoiding catastrophic forgetting of pre-trained foundation models and gradually integrates exogenous information. Extensive experiments show that CoRA of TSFMs surpasses state-of-the-art covariate-aware deep forecasters with full or few-shot training samples, achieving 31.1% MSE reduction on covariate-aware forecasting. Compared to other adaptation methods, CoRA exhibits strong compatibility with various advanced TSFMs and extends the scope of covariates to other modalities, presenting a practical paradigm for the application of TSFMs.

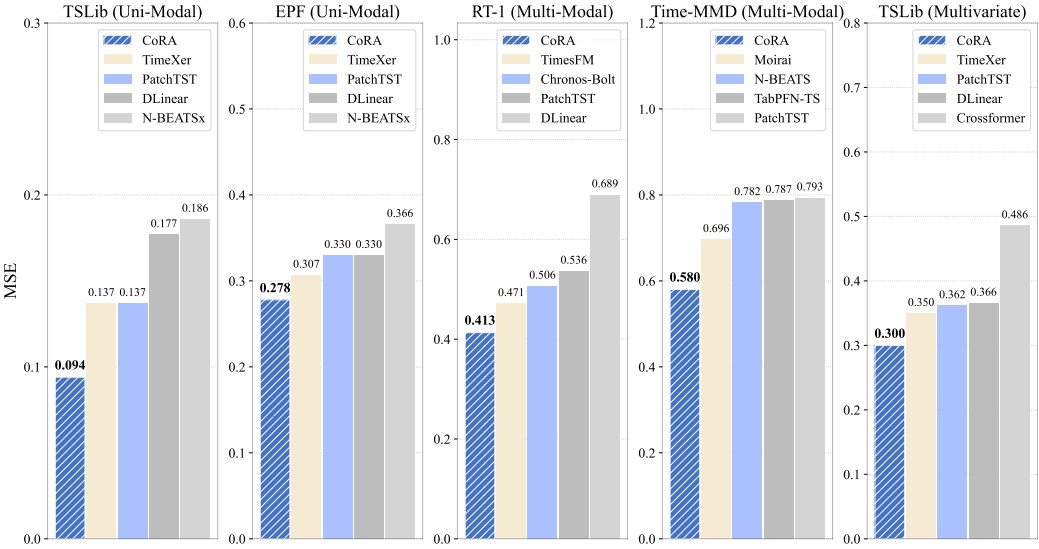

Figure 1: CoRA performance on different covariate-aware benchmarks.

# 1 INTRODUCTION

Time series forecasting has gained increasing prominence in real-world applications, such as weather forecasting (Hittawe et al., 2024), supply chain optimization (Panda & Mohanty, 2023) and financial market assessment (Cheng et al., 2022). With the rapid development of large-scale time-series datasets (Woo et al., 2023) and scalable architectures (Vaswani et al., 2017), recent research has focused on developing Time Series Foundation Models (TSFMs) (Das et al., 2023b; Liu et al., 2024c; Ansari et al., 2024; Liu et al., 2025), which exhibit impressive scalability and out-of-box generalization performance across various applications.

Despite time series are typically multi-dimensional data, most TSFMs are pre-trained on univariate time series (Das et al., 2023b; Liu et al., 2024d; Shi et al., 2024), primarily due to the considerable heterogeneity in dimensionality and inter-variate relationships across datasets. In particular, the dependencies among variates in one dataset often fail to generalize to others. For example, transferring relationships learned from meteorological variates to the financial domain may not be sensible. Besides, covariate-aware deep forecasters, which are trained in a channel-dependence approach (Qiu et al., 2025), have not been well-demonstrated to be scalable and versatile. Meanwhile, an important paradigm of foundation models involves large-scale pre-training on general large-scale data and adaptation to task-specific datasets. Therefore, these constraints necessitate the paradigm shift as shown in Figure 2, which adapts TSFMs to covariate-aware forecasting scenarios while revitalizing the pre-trained backbone of foundation models (Arango et al., 2025; Benechehab et al., 2025).

Different from adaptation methods for language models such as LoRA (Hu et al., 2021), covariate-aware adaptation in time series forecasting faces fundamentally different challenges. The difficulty lies in the multi-dimensionality and the heterogeneity of modalities in covariates. Simply incorporating exogenous information into the target variate is insufficient, because dependencies among variates are often domain-specific, noncausal, and sometimes noisy. Therefore, adaptation of TSFM requires not only the integration of covariate information but also evaluating the causality of different covariates. Guided by the principled criteria, we delve into Granger causality, a foundational concept for identifying causal dependencies in time series forecasting (Granger, 1969), and develop a date-dependent approach to ground covariate-aware adaptation with interpretable modular design.

While prior works (Arango et al., 2025; Benechehab et al., 2025; Han et al., 2025) attempt to incorporate time series covariates into TSFMs, they inject covariate-aware modules that alter the embeddings away from the pre-trained embedding space. Besides, previous adaptation methods introduce trainable modules without zero-initialization, implying that the initial outputs of the adapted model are no longer equivalent to the pre-trained TSFMs. Empirically, adaptation without zero-initialization will cause unstable training, catastrophic forgetting and sometimes even worse performance than just zero-shot evaluation (Hu et al., 2021; Peebles & Xie, 2023).

In this paper, we introduce **CoRA**, a general, effective, and interpretable framework to adapt TSFMs on covariate-aware forecasting tasks, where covariates cover time series, language, images, and other structured data. Concretely, CoRA treats pre-trained foundation models of different modalities as frozen embedding extractors. With extracted embeddings from raw covariates, CoRA includes a covariate evaluation and routing module, termed Causality Embedding, which automatically produces a causally-informed significance score during adaptation. These embeddings are then integrated

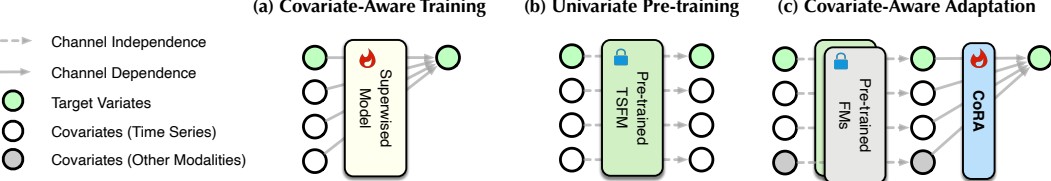

Figure 2: Several paradigms of time series forecasting: (a) Covariate-aware deep models are supervisedly trained in a channel-dependent way. However, the backbone can be task-specific and challenged to scale up. (b) TSFMs designed to address data heterogeneity are generally pre-trained and predict on univariate time series. which makes them infeasible to utilize inter-variate dependencies explicitly. (c) CoRA leverages various foundation models, incorporates exogenous information to predict the target variate, and rapidly adapts to specific tasks without altering pre-trained models.

through a zero-initialized condition-injection mechanism by learning scale and shift parameters. CoRA achieves state-of-the-art performance while requiring fewer samples compared to supervised models and previous adaptation methods. In-depth studies validate the generality and interpretability of the proposed framework. Our main contributions are summarized as follows:

- We emphasize that an important paradigm of covariate-aware forecasting on TSFMs, which effectively revitalize pre-trained foundation models and address the unique challenges in utilizing high-dimensional, multi-modal, and causally-dependent covariates.
- We propose CoRA, a general and effective covariate-aware adaptation framework that freezes pre-trained models and introduces a Causality Embedding for principled covariate selection, combined with a zero-initialized condition-injection mechanism.
- Extensive experiments across diverse benchmarks demonstrate that CoRA achieves state-of-the-art performance, requires fewer training samples, and provides interpretable insights into covariate causality, surpassing both supervised models and other adaptation methods.

## 2 RELATED WORK

### 2.1 TIME SERIES FOUNDATION MODELS

Recent research has explored pre-training Time Series Foundation Models (TSFMs) on large-scale datasets, enabling strong zero-shot generalization to downstream tasks. TimesFM (Das et al., 2023b) and Timer (Liu et al., 2024d) are the first to adopt a decoder-only Transformer architecture with the next-token prediction objective. Chronos (Ansari et al., 2024) introduces a discretization approach for time series and predicts next tokens using LLM backbone and language modeling. Sundial (Liu et al., 2025) proposes TimeFlow, incorporating generative modeling to realize the flexibility of probabilistic forecasting. However, these models are limited to univariate pre-training, which restricts their applicability to downstream tasks involving multi-dimensional or multi-modal covariates. One exception is that Moirai (Woo et al., 2024) adopts multivariate pre-training by flattening variates and appending variate-wise embeddings, but it has to subsample multivariate series with a fixed size for training stability, leading to incomplete perception for high-dimensional time series inputs.

### 2.2 COVARIATE-AWARE DEEP FORECASTERS

In real-world time series forecasting, covariates play a crucial role in improving the predictability of target variate. Classical approaches such as ARIMAX (Williams, 2001) and SARIMAX (Vagropoulos et al., 2016) model the correlations between covariates and the target variate by linear regression. More recent deep learning methods, such as the Temporal Fusion Transformer (Lim et al., 2021), emphasize variate selection as a key mechanism. Other approaches, including NBEATSx (Olivares et al., 2023) and TiDE (Das et al., 2023a), argue that forecasting models can directly leverage future covariate information when predicting target values. TimeXer (Wang et al., 2024) achieves competent performance by modeling the target variate at the patch level and the covariates at the series level. Time-VLM (Zhong et al., 2025) leverages vision-language backbones to integrate temporal, visual, and textual information for multi-modal forecasting. However, supervised deep models trained from scratch may yield suboptimal performance without substantial task-specific data.

### 2.3 ADAPATION METHODS OF FOUNDATION MODELS

Adaptation of foundation models such as LoRA (Hu et al., 2021; Dettmers et al., 2023) is typically applied in language and vision models, where the upstream and downstream tasks share the same 1D-sequence structure. In contrast, adapting univariate pre-trained TSFMs to covariate-aware scenarios introduces dimensional changes in the input structure. Prior works such as ChronosX (Arango et al., 2025), AdaPTS (Benechehab et al., 2025), and UniCA (Han et al., 2025) modify the TSFM input structure by injecting covariates before the backbone, which inevitably alters the pre-trained embedding space and may trigger catastrophic forgetting. In contrast, Gen-P-Tuning (Liu et al., 2024b) learns covariate prompts at the front of the context, introducing a relatively smaller structural change. Moreover, adaptation of foundation models relies on zero-initialization (Goyal et al., 2017) to ensure that the training start-point begins consistently with the pre-trained model. However, such principled strategies have not been properly considered in existing TSFMs adaptation methods.

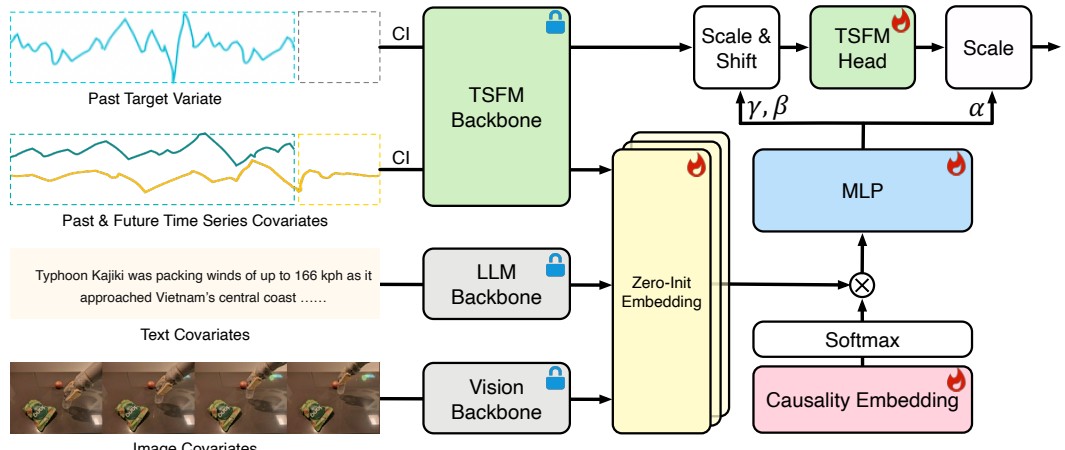

Figure 3: Overall architecture of CoRA. CoRA freezes the backbone of foundation models as embedding extractors for multi-modal covariates, which are then selected by a trainable Causality Embedding. This refined embedding is injected into the original TSFM head via a zero-initialized module to generate the shifting and scaling factors for final predictions.

## 3 APPROACH

In covariate-aware forecasting, we consider one target variate $\mathbf{x}_{1:T} = \{x_1, \ldots, x_T\} \in \mathbb{R}^T$ observed over $T$ time steps along with exogenous covariates $\mathbf{C}_{1:\tau} = \{\mathbf{C}_1, \ldots, \mathbf{C}_\tau\}$ [1]. The task is to train a forecaster $f_\theta$ parameterized by $\theta$ that can predict the target variate $\mathbf{x}_{T+1:T+H} = \{x_{T+1}, \ldots, x_{T+H}\}$ for the next $H$ time steps:

$$f_\theta : (\mathbf{x}_{1:T}, \mathbf{C}_{1:\tau}) \mapsto \hat{\mathbf{x}}_{T+1:T+H}. \tag{1}$$

### 3.1 FOUNDATION MODELS AS FROZEN EMBEDDING EXTRACTOR

For real-world forecasting, exogenous covariates are very often multi-dimensional (e.g., multivariate time series) and multi-modal. In contrast to previous methods that solely adapt the foundation model of time series, we categorize exogenous covariates into three mainstream modalities. As illustrated in Figure 3, we separate covariates as $N$ one-dimensional sequences, such as univariate time series, text, or image snapshots, and extract per-step embeddings from corresponding frozen models:

$$\mathbf{E}_{1:\tau_i}^{m_i} = \text{FM-Backbone}(\mathbf{C}_{1:\tau_i}^{m_i}), \ i = 1, \ldots, N, \ m_i \in \{\text{ts}, \text{txt}, \text{img}\}. \tag{2}$$

At each time step, the embeddings $\mathbf{E}_t^{\text{ts}} \in \mathbb{R}^{N_{\text{ts}} \times D_{\text{ts}}}$, $\mathbf{E}_t^{\text{txt}} \in \mathbb{R}^{N_{\text{txt}} \times D_{\text{txt}}}$, and $\mathbf{E}_t^{\text{img}} \in \mathbb{R}^{N_{\text{img}} \times D_{\text{img}}}$ capture the exogenous information of corresponding covariates by leveraging the embeddings generated before the last layer of the foundation models, where $D_{\text{ts}}, D_{\text{txt}}, D_{\text{img}}$ denote the latent dimensions of the respective foundation models and $N_{\text{ts}}, N_{\text{txt}}, N_{\text{img}}$ represent the number of covariates categorized into each modality, with the total number of covariates $N = N_{\text{ts}} + N_{\text{txt}} + N_{\text{img}}$.

For dynamic covariates that are recorded at each time step, CoRA regards one covariate as a whole by aggregating the embeddings over all time steps. For typical TSFMs adopting the decoder-only or encoder-decoder architecture, we employ the last-step embedding that corresponds to the latest-known values, which captures all previous context in one single-series covariate. For language and vision foundation models that encode one snapshot, we utilize the averaged embeddings across all snapshots of time steps (for simplicity, we omit the variate index $i$):

$$\tilde{\mathbf{E}}^{\text{ts}} = \mathbf{E}_\tau^{\text{ts}}, \ \tilde{\mathbf{E}}^{\text{txt}} = \frac{1}{\tau} \sum_{t=1}^{\tau} \mathbf{E}_t^{\text{txt}}, \ \tilde{\mathbf{E}}^{\text{img}} = \frac{1}{\tau} \sum_{t=1}^{\tau} \mathbf{E}_t^{\text{img}}. \tag{3}$$

For the target variate, we use the TSFM backbone to extract its embeddings and take the embedding at the last time step $T$ to capture the overall lookback information:

$$\mathbf{E}_{1:T}^{\text{target}} = \text{TSFM-Backbone}(\mathbf{x}_{1:T}), \ \tilde{\mathbf{E}}^{\text{target}} = \mathbf{E}_T^{\text{target}}. \tag{4}$$

[1]Covariates may be future-unknown ($\tau = T$), future-known ($\tau = T + H$), or static covariates ($\tau = 1$).

## 3.2 COVARIATE-AWARE ADAPTATION

**Granger Causality**   Granger causality test (Granger, 1969) is a statistical hypothesis test used to determine whether using a covariate $\mathbf{C}$ and $\mathbf{x}_{1:T}$ to predict $\mathbf{x}_{T+1:T+H}$ yields a lower prediction error than using $\mathbf{x}_{1:T}$ alone. If so, $\mathbf{C}$ is said to Granger causes $\mathbf{x}$. Unlike real-world "who-causes-whom" causal relationships, Granger causality captures the predictive usefulness of $C$ for forecasting $\mathbf{x}_{T+1:T+H}$, not whether $C$ is the true causal driver of $\mathbf{x}_{T+1:T+H}$. A covariate can aid prediction without directly causing $\mathbf{x}_{T+1:T+H}$. For example, if a latent variable $y$ causes both $C$ and $\mathbf{x}_{T+1:T+H}$, $C$ may still improve the prediction of $\mathbf{x}$, and thus be regarded as a Granger cause of $\mathbf{x}_{T+1:T+H}$. Granger causality also differs from simple correlations. For example, a sine and cosine wave have zero correlation, yet the Granger causality test between them can be significant.

**Covariate Selection**   In typical covariate-aware forecasting tasks, multiple covariates are involved, and their significance of Granger causality with respect to the target variate may differ considerably. Therefore, we introduce a trainable Causality Embedding $\mathbf{W}_{\text{CE}} \in \mathbb{R}^N$, which learns to quantify the causal influence of each covariate on $\mathbf{x}_{1:T}$. Empirically, we observe that the learned Causality Embedding exhibits highly consistent result with the statistical test of Granger causality in Section 4.2. Concretely, we first align the embeddings of multi-modal covariates into a unified hidden space since the latent dimensions of foundation models are not necessarily identical:

$$\hat{\mathbf{E}}^{m_i} = \tilde{\mathbf{E}}^{m_i} \mathbf{W}^{m_i} + \mathbf{b}^{m_i}, \; i = 1, \ldots, N, \; m_i \in \{\text{ts}, \text{txt}, \text{img}\},$$
$$\hat{\mathbf{E}} = \text{Concat}\left(\hat{\mathbf{E}}^{\text{ts}}, \hat{\mathbf{E}}^{\text{txt}}, \hat{\mathbf{E}}^{\text{img}}\right). \tag{5}$$

where $\mathbf{W}^{m_i} \in \mathbb{R}^{D_{m_i} \times D}$, $\mathbf{b}^{m_i} \in \mathbb{R}^D$ for $m_i \in \{\text{ts}, \text{txt}, \text{img}\}$, and $\hat{\mathbf{E}} \in \mathbb{R}^{N \times D}$. Afterwards, we use Causality Embedding $W_{\text{CE}} \in \mathbb{R}^N$ to evaluate and gate each covariate during the adaptation process, yielding a unified embedding that aligns the latent space of TSFMs:

$$\mathbf{H} = \text{Softmax}(\mathbf{W}_{\text{CE}}) \cdot \hat{\mathbf{E}}. \tag{6}$$

**Covariate Injection**   With obtained overall exogenous embeddings of all covariates, we adopt an adaptive layer-normalization (adaLN) layer proposed by DiT (Peebles & Xie, 2023), which is widely shown to outperform approaches such as concatenation and cross-attention on continuous-valued modality. Specifically, $\mathbf{H}$ is mapped into $\alpha \in \mathbb{R}^H$ and $\beta, \gamma \in \mathbb{R}^D$ via a lightweight $\text{MLP}(\cdot)$. The outcomes are then applied via shift-and-scale operations to modulate the statistics before and after the original head of TSFM, thereby injecting the covariate information into the adaptation process. Finally, we adopt the identical loss function used in the pre-trained TSFM for training:

$$\gamma, \beta, \alpha = \text{MLP}\left(\mathbf{H}\right),$$
$$\hat{\mathbf{x}}_{T+1:T+H} = (1 + \alpha) \, \text{TSFM-Head}\left(\gamma + (1 + \beta) \, \tilde{\mathbf{E}}^{\text{target}}\right). \tag{7}$$
$$\text{loss} = \text{TSFM-Loss}(\hat{\mathbf{x}}_{T+1:T+H}, \mathbf{x}_{T+1:T+H})$$

**Zero-Initialization**   Similar to LoRA (Hu et al., 2021), we zero-initialize the parameters of $\mathbf{W}^{m_i} \in \mathbb{R}^{D_{m_i} \times D}$, $\mathbf{b}^{m_i} \in \mathbb{R}^D$ for $m_i \in \{\text{ts}, \text{txt}, \text{img}\}$ and the MLP. Therefore, the overall model is identical to the pre-trained TSFM. This design ensures adaptation begins from the pre-trained state, while progressively integrating additional information in a stable and incremental manner.

## 4 EXPERIMENTS

We conduct comprehensive experiments to evaluate the effectiveness of CoRA, covering uni-modal and multi-modal covariate-aware forecasting, few-shot forecasting, and extensions to multivariate forecasting. The overall performance is provided in Figure 1. We further provide in-depth analysis, including generality across different TSFMs, ablation studies, and model interpretability.

## 4.1 MAIN RESULTS

In this section, we conduct extensive experiments to evaluate the performance of CoRA, compared with existing adaptation methods and advanced supervised deep forecasters. For fair comparison, we adopt Sundial (Liu et al., 2025) as the backbone model for all adaptation approaches. Moreover, we ensure none of the test sets overlap with Sundial's training data to avoid potential data leakage.

Table 1: Averaged results of the long-term covariate-aware forecasting. For all baselines, the look-back length $L$ is fixed at 2880. The reported performance is averaged over prediction horizons $S = \{96, 192, 336, 720\}$ and full results are provided in Table 8. Dash (-) denotes out of memory.

| Models | CoRA (Ours) | | AdaPTS (2025) | | ChronosX (2025) | | UniCA (2025) | | TimeXer (2024) | | iTransformer (2023) | | PatchTST (2022) | | NBEATSx (2023) | | Crossformer (2023) | | DLinear (2023) | |
|---|---|---|---|---|---|---|---|---|---|---|---|---|---|---|---|---|---|---|---|---|
| Metric | MSE | MAE | MSE | MAE | MSE | MAE | MSE | MAE | MSE | MAE | MSE | MAE | MSE | MAE | MSE | MAE | MSE | MAE | MSE | MAE |
| ETTh1 | 0.068 | 0.203 | 0.076 | 0.211 | 0.085 | 0.227 | 0.085 | 0.222 | 0.089 | 0.240 | 0.160 | 0.317 | 0.096 | 0.249 | 0.181 | 0.351 | 0.386 | 0.501 | 0.263 | 0.408 |
| ETTh2 | 0.141 | 0.299 | 0.156 | 0.311 | 0.365 | 0.466 | 0.197 | 0.350 | 0.194 | 0.355 | 0.307 | 0.445 | 0.191 | 0.352 | 0.181 | 0.351 | 0.395 | 0.502 | 0.320 | 0.454 |
| ETTm1 | 0.043 | 0.155 | 0.046 | 0.165 | 0.049 | 0.165 | 0.050 | 0.166 | 0.062 | 0.192 | 0.059 | 0.186 | 0.055 | 0.181 | 0.112 | 0.268 | 0.068 | 0.207 | 0.059 | 0.184 |
| ETTm2 | 0.100 | 0.237 | 0.107 | 0.245 | 0.106 | 0.246 | 0.122 | 0.265 | 0.161 | 0.304 | 0.149 | 0.304 | 0.131 | 0.278 | 0.222 | 0.384 | 0.208 | 0.366 | 0.123 | 0.266 |
| Weather | 0.001 | 0.026 | 0.002 | 0.027 | 0.002 | 0.033 | 0.002 | 0.033 | 0.002 | 0.033 | 0.002 | 0.034 | 0.002 | 0.036 | 0.033 | 0.086 | 0.004 | 0.047 | 0.008 | 0.076 |
| ECL | 0.194 | 0.314 | 0.212 | 0.329 | 0.206 | 0.323 | 0.230 | 0.347 | 0.292 | 0.387 | 0.293 | 0.406 | 0.327 | 0.431 | 0.352 | 0.449 | 0.352 | 0.446 | 0.264 | 0.376 |
| Traffic | 0.112 | 0.186 | - | - | - | - | 0.122 | 0.203 | 0.157 | 0.259 | 0.139 | 0.232 | 0.154 | 0.255 | 0.222 | 0.328 | 0.274 | 0.332 | 0.203 | 0.317 |

Table 2: Full results of the short-term covariate-aware forecasting. Following the standard protocol of EPF dataset, with input-output lengths of 168-24. Avg means the average results from all five datasets. Results of end-to-end models are officially reported by TimeXer (Wang et al., 2024).

| Models | CoRA (Ours) | | AdaPTS (2025) | | UniCA (2025) | | ChronosX (2025) | | TimeXer (2024) | | iTransformer (2023) | | PatchTST (2022) | | NBEATSx (2023) | | Crossformer (2023) | | DLinear (2023) | |
|---|---|---|---|---|---|---|---|---|---|---|---|---|---|---|---|---|---|---|---|---|
| Metric | MSE | MAE | MSE | MAE | MSE | MAE | MSE | MAE | MSE | MAE | MSE | MAE | MSE | MAE | MSE | MAE | MSE | MAE | MSE | MAE |
| NP | 0.222 | 0.246 | 0.231 | 0.259 | 0.265 | 0.289 | 0.254 | 0.278 | 0.236 | 0.268 | 0.265 | 0.300 | 0.267 | 0.284 | 0.272 | 0.301 | 0.240 | 0.285 | 0.309 | 0.321 |
| PJM | 0.073 | 0.165 | 0.080 | 0.173 | 0.090 | 0.187 | 0.089 | 0.189 | 0.093 | 0.192 | 0.097 | 0.197 | 0.106 | 0.209 | 0.097 | 0.189 | 0.101 | 0.199 | 0.108 | 0.215 |
| BE | 0.339 | 0.236 | 0.355 | 0.261 | 0.368 | 0.273 | 0.371 | 0.274 | 0.379 | 0.243 | 0.394 | 0.270 | 0.400 | 0.262 | 0.389 | 0.265 | 0.420 | 0.290 | 0.463 | 0.313 |
| FR | 0.357 | 0.206 | 0.363 | 0.218 | 0.365 | 0.218 | 0.361 | 0.217 | 0.385 | 0.208 | 0.439 | 0.233 | 0.411 | 0.220 | 0.393 | 0.211 | 0.434 | 0.208 | 0.429 | 0.260 |
| DE | 0.401 | 0.388 | 0.455 | 0.424 | 0.553 | 0.466 | 0.453 | 0.426 | 0.440 | 0.415 | 0.479 | 0.443 | 0.461 | 0.432 | 0.499 | 0.447 | 0.574 | 0.430 | 0.520 | 0.463 |
| AVG | 0.278 | 0.248 | 0.297 | 0.267 | 0.328 | 0.287 | 0.306 | 0.277 | 0.307 | 0.265 | 0.335 | 0.289 | 0.330 | 0.282 | 0.330 | 0.283 | 0.354 | 0.284 | 0.366 | 0.314 |

### 4.1.1 UNI-MODAL COVARIATE-AWARE FORECASTING

**Setups** In the uni-modal setting, all covariates are time series. We conduct both long-term and short-term uni-modal covariate-aware forecasting experiments. In the long-term setting, we use seven real-world datasets, including ECL, ETT (4 subsets), Traffic, and Weather, employed in Autoformer (Wu et al., 2021), where the final dimension serves as the target variate and the remaining dimensions as covariates. In the short-term setting, we adopt the electricity price forecasting (EPF) task (Lago et al., 2021), with electricity price as the target variate and two correlated covariates.

**Results** As shown in Table 1 and Table 2, CoRA delivers state-of-the-art performance across both long- and short-term forecasting. Specifically, in long-term forecasting, CoRA outperforms the strongest supervised model TimeXer (Wang et al., 2024), by 31.1% in MSE and 19.8% in MAE, stressing the advantage of building on pre-trained TSFMs rather than training task-specific models from scratch. Compared to other adaptation methods, using the same model Sundial (Liu et al., 2025), CoRA reduces MSE by 18.7% compared to the second best adaptation method UniCA (Han et al., 2025), highlighting the importance of maintaining parameter consistency and equivalent initialization during adaptation. In the EPF task, CoRA reduces MSE by 9.4% compared to TimeXer and by 6.4% compared to AdaPTS (Benechehab et al., 2025), further solidifying its position as a superior and generalized approach for uni-modal covariate-aware forecasting.

### 4.1.2 MULTI-MODAL COVARIATE-AWARE FORECASTING

**Setups** We evaluate CoRA on tasks involving multi-modal covariates, specifically images and text. For image-based covariates, we construct a subset from the RT-1 (Brohan et al., 2022) dataset, which contains a target time series with image covariates at each timestamp. For text-based covariates, we choose the Time-MMD (Liu et al., 2024a) dataset, which includes a target time series

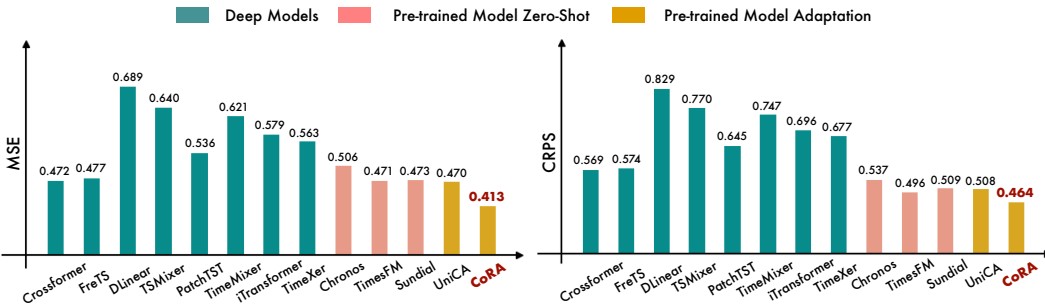

Figure 4: Multi-modal covariate-aware forecasting on a subset of RT-1 (Brohan et al., 2022) with a time series target variate and an image covariate. Input length is set to 32 and prediction length is 4.

Table 3: Multi-modal covariate-aware forecasting on Time-MMD (**?**) with textual covariates. Baseline results are reported by UniCA (Han et al., 2025), with full results in Table 9.

| Models | CoRA (Ours) | UniCA (2025) | Sundial (2025) | Moirai (2024) | TabPFN-TS (2025) | PatchTST (2022) | TTM (2024) | TiDE (2023a) | N-BEATS (2023) | TFT (2021) | DeepAR (2020) |
|---|---|---|---|---|---|---|---|---|---|---|---|
| Average | **0.641** | 0.661 | 0.662 | 0.751 | 0.795 | 0.933 | 0.820 | 0.927 | 0.882 | 0.947 | 1.361 |
| MSE | **0.580** | 0.591 | 0.591 | 0.696 | 0.787 | 0.793 | 0.685 | 0.869 | 0.782 | 0.992 | 1.605 |
| MAE | **0.690** | 0.716 | 0.716 | 0.821 | 0.837 | 1.009 | 0.866 | 0.976 | 0.884 | 0.958 | 1.219 |
| CRPS | **0.653** | 0.677 | 0.678 | 0.735 | 0.762 | 0.996 | 0.909 | 0.937 | 0.980 | 0.891 | 1.260 |

with a corresponding text covariate. Moreover, CoRA adopts ViT[2] (Wu et al., 2020) and Qwen3-Embedding[3] (Zhang et al., 2025) as backbone to extract features from image and text respectively.

**Results** As shown in Figure 4 and Table 3, CoRA achieves state-of-the-art performance across all metrics. On the RT-1 (**?**) dataset, CoRA outperforms the best end-to-end supervised model and TSFM zero-shot by 12.7% in MSE and 8.8% in CRPS. While on the Time-MMD benchmark (**?**), the improvements are 1.9% in MSE and 3.7% in CRPS. These results demonstrate that properly modeling auxiliary modalities provides substantial benefits for forecasting. Compared with UniCA (Han et al., 2025), which does not maintain backbone consistency or use proper zero-initialization, CoRA consistently achieves superior performance on both benchmarks.

### 4.1.3 FEW-SHOT FORECASTING

**Setups** In real-world applications, the available training data is often highly limited, making few-shot forecasting a critical challenge for robust deployment. We evaluate CoRA on the well-established electricity price forecasting (EPF) task (Lago et al., 2021), comparing it with alternative adaptation methods and end-to-end models across a range of data scarcity levels.

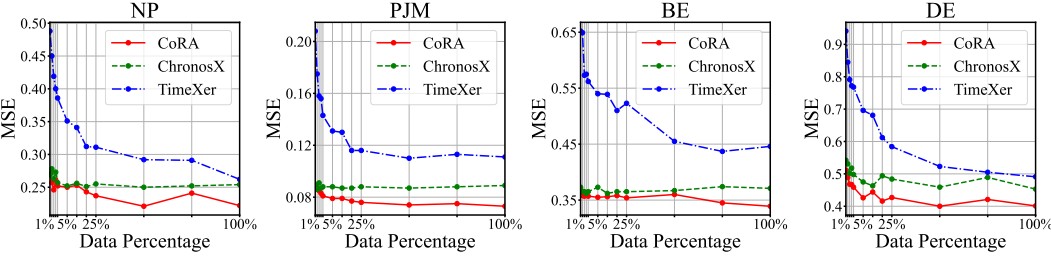

Figure 5: Few-shot forecasting on the EPF dataset, comparing CoRA with TimeXer (Wang et al., 2024) and ChronosX (Arango et al., 2025) across different levels of data availability.

**Results** As shown in Figure 5, CoRA consistently outperforms TimeXer (Wang et al., 2024) and ChronosX (Arango et al., 2025) under different data availability levels. When the number of samples

---

[2]https://huggingface.co/google/vit-base-patch16-224-in21k.
[3]https://huggingface.co/Qwen/Qwen3-Embedding-0.6B.

is particularly small (1% to 25%), the end-to-end model TimeXer performs significantly worse than adaptation methods based on pre-trained TSFMs, highlighting that pre-trained models can adapt to downstream tasks more quickly and effectively with limited data. Even with sufficient data, TimeXer still underperforms compared with adaptation methods, due to its relatively smaller model capacity. Moreover, thanks to principled designs that preserve the pre-trained backbone and employ proper zero-initialization, CoRA consistently outperforms ChronosX.

### 4.1.4 Multivariate Time Series Forecasting

**Setups** CoRA naturally extends to the multivariate time series forecasting scenarios via the channel-independence mechanism, enabling joint prediction of multiple target variates. We evaluate this on seven real-world datasets introduced in Autoformer (Wu et al., 2021).

**Results** As shown in Table 4, CoRA outperforms all other supervised forecasters, achieving average MSE and MAE reductions of 14.5% and 12.2% compared to TimeXer (Wang et al., 2024). CoRA's superior performance stems from its use of pre-trained TSFMs that have already internalized universal temporal patterns from large-scale datasets. This enables CoRA to more accurately capture inter-variate dependencies and generalize effectively across diverse datasets.

Table 4: Averaged results of the multivariate forecasting task on well-acknowledged benchmarks. For all baselines, the look-back length $L$ is fixed at 2880. The reported performance is averaged over prediction horizons $S = \{96, 192, 336, 720\}$ and full results are provided in Table 10.

| Models | CoRA (Ours) | | Timer-XL (2024d) | | TimeXer (2024) | | iTransformer (2023) | | PatchTST (2022) | | Crossformer (2023) | | TiDE (2023a) | | DLinear (2023) | | SCINet (2022) | | Autoformer (2021) | |
|---|---|---|---|---|---|---|---|---|---|---|---|---|---|---|---|---|---|---|---|---|
| Metric | MSE | MAE | MSE | MAE | MSE | MAE | MSE | MAE | MSE | MAE | MSE | MAE | MSE | MAE | MSE | MAE | MSE | MAE | MSE | MAE |
| ETTh1 | **0.404** | **0.422** | 0.548 | 0.547 | 0.492 | 0.488 | 0.508 | 0.515 | 0.516 | 0.504 | 0.643 | 0.594 | 0.656 | 0.587 | 0.519 | 0.512 | 0.780 | 0.660 | 0.812 | 0.661 |
| ETTh2 | **0.331** | **0.381** | 0.422 | 0.454 | 0.454 | 0.476 | 0.440 | 0.476 | 0.490 | 0.503 | 0.810 | 0.691 | 0.555 | 0.532 | 0.620 | 0.589 | 0.667 | 0.592 | 0.840 | 0.707 |
| ETTm1 | **0.337** | **0.371** | 0.381 | 0.419 | 0.398 | 0.424 | 0.379 | 0.413 | 0.400 | 0.424 | 0.436 | 0.457 | 0.363 | 0.393 | 0.357 | 0.387 | 0.425 | 0.447 | 0.857 | 0.682 |
| ETTm2 | **0.256** | **0.317** | 0.318 | 0.383 | 0.274 | 0.343 | 0.276 | 0.342 | 0.292 | 0.355 | 0.569 | 0.593 | 0.306 | 0.370 | 0.266 | 0.335 | 0.308 | 0.378 | 0.457 | 0.495 |
| Weather | **0.230** | **0.269** | 0.316 | 0.348 | 0.262 | 0.303 | 0.251 | 0.305 | 0.251 | 0.290 | 0.235 | 0.285 | 0.234 | 0.281 | 0.237 | 0.291 | 0.249 | 0.296 | 0.500 | 0.487 |
| ECL | **0.155** | **0.250** | 0.155 | 0.252 | 0.172 | 0.275 | 0.194 | 0.299 | 0.163 | 0.265 | 0.184 | 0.281 | 0.160 | 0.254 | 0.156 | 0.255 | 0.181 | 0.285 | 0.292 | 0.390 |
| Traffic | **0.384** | **0.265** | 0.597 | 0.510 | 0.401 | 0.281 | 0.407 | 0.291 | 0.422 | 0.298 | 0.522 | 0.285 | 0.402 | 0.276 | 0.406 | 0.284 | 0.478 | 0.352 | 0.742 | 0.464 |

### 4.2 Model Analysis

In this section, we perform thorough experiments to analyze several properties of CoRA, including its generalization to other TSFMs such as TimesFM (Das et al., 2023b), Chronos-bolt (Ansari et al., 2024), and FlowState (Graf et al., 2025), ablation studies on the method's key components, and the interpretability of learned Causality Embedding.

**Generality** Figure 6 shows that CoRA further boosts the performance of various TSFMs on top of their zero-shot results. Average MSE reductions are 14.2% on Sundial (Liu et al., 2025), 3.3% on TimesFM (Das et al., 2024), 4.9% on Chronos-Bolt (Ansari et al., 2024), and 3.3% on FlowState (Graf et al., 2025). These results demonstrate that CoRA offers an effective and flexible adaptation strategy, seamlessly integrating with diverse backbone architectures.

**Ablation Study** We provide a thorough ablation study to examine our proposed CoRA in Table 5. Our results show that each component is crucial for CoRA's performance by addressing specific challenges in covariate-aware time series forecasting. Without the covariates' information, forecasting performance degrades, underscoring the necessity of incorporating external signals to enhance the predictability of the target. Without the adaLN module, we find that simply adding the condition to the TSFM head input is insufficient. Instead, our condition-injection mechanism is highly effective by influencing the statistics of the TSFM head to fuse information. Similarly, when we removed the Causality Embedding, replacing it with mean aggregation, the model's performance dropped. This demonstrates the importance of our selection and routing mechanism, which automatically assigns appropriate weights to different covariates based on their inherent causality. Finally, we

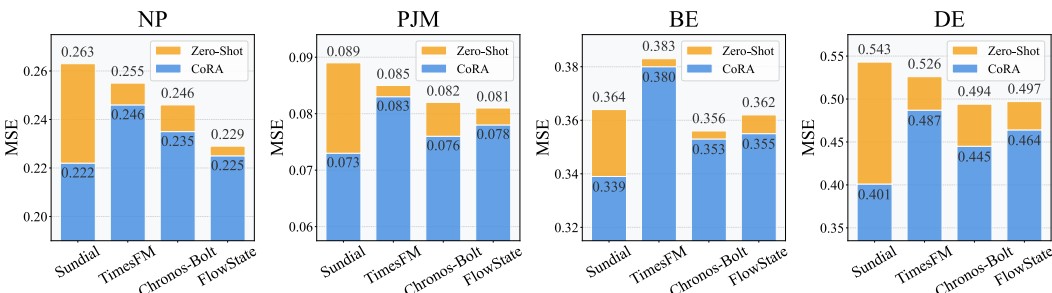

Figure 6: Performance gains of CoRA across diverse TSFMs. Full results are provided in Table 11.

observed that replacing zero-initialization with Xavier initialization resulted in worse performance. This confirms that zero-initialization is vital for preserving the valuable knowledge learned during pre-training and ensuring a stable adaptation process.

Table 5: Ablation study of CoRA. (1) *w/o* covariate denotes Supervised Fine-Tuning (SFT), trained without using covariates. (2) *w/o* adaLN replaces the adaLN module by directly adding the condition to the input of the TSFM head. (3) *w/o* selection replaces the Causality Embedding with mean aggregation. (4) *w/o* zero-init replaces zero-initialization with Xavier initialization.

| Datasets | NP | | PJM | | BE | | FR | | DE | | Avg | |
|---|---|---|---|---|---|---|---|---|---|---|---|---|
| Models | MSE | MAE | MSE | MAE | MSE | MAE | MSE | MAE | MSE | MAE | MSE | MAE |
| **CoRA** | **0.222** | **0.246** | **0.073** | **0.165** | **0.339** | **0.236** | **0.357** | **0.206** | **0.401** | **0.388** | **0.278** | **0.248** |
| w/o covariate | 0.231 | 0.256 | 0.078 | 0.172 | 0.352 | 0.262 | 0.360 | 0.214 | 0.458 | 0.426 | 0.296 | 0.266 |
| w/o adaLN | 0.260 | 0.288 | 0.085 | 0.180 | 0.351 | 0.238 | 0.368 | 0.210 | 0.506 | 0.451 | 0.314 | 0.273 |
| w/o selection | 0.273 | 0.266 | 0.080 | 0.177 | 0.356 | 0.262 | 0.360 | 0.215 | 0.472 | 0.423 | 0.301 | 0.269 |
| w/o zero-init | 0.234 | 0.262 | 0.078 | 0.173 | 0.350 | 0.257 | 0.360 | 0.208 | 0.430 | 0.415 | 0.290 | 0.263 |

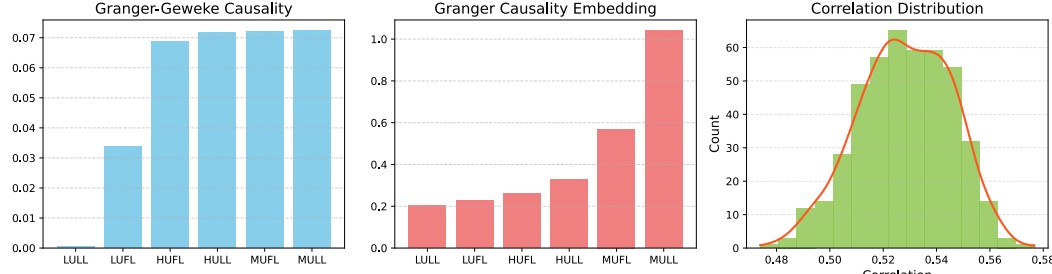

Figure 7: Correlation between traditional statistic Granger-Geweke Causality (Dhamala et al., 2018) and the Causality Embedding learned in CoRA on ETTh1 Dataset.

**Interpretability** To study the interpretability of CoRA, we compare the learned Causality Embedding with the traditional Granger-Geweke Causality (Dhamala et al., 2018). We select 1000 windows from the ETTh1 dataset and compute the Granger-Geweke Causality for each window (detailed description in the Algorithm 2) as well as the Causality Embedding learned by CoRA. Figure 7 demonstrates a strong correlation between the Granger–Geweke Causality and the Causality Embedding. Furthermore, we plot a histogram of the Pearson correlation coefficient (Pearson, 1895) across the 1000 windows, which clearly demonstrates their consistency.

## 5 CONCLUSION

In this paper, we introduce CoRA, a general, flexible, and interpretable framework for adapting pre-trained foundation models to covariate-aware forecasting tasks. An important paradigm of foundation models involves large-scale pre-training on general datasets followed by adaptation to task-

specific datasets. CoRA leverages this paradigm by using the powerful backbones of diverse foundation models as frozen embedding extractors. It then employs a Causality Embedding to weight and select covariates based on their causal relationship to the target variate, and a zero-initialized adaLN module for stable and progressive fusion of this information. Our extensive experiments consistently show that CoRA outperforms both advanced supervised models and other adaptation methods while requiring fewer training samples, bridging the gap between powerful pre-trained models and the complex multi-modal and multivariate challenges of real-world scenarios.

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

## A  EXPERIMENTAL DETAILS

### A.1  DATASETS

To comprehensively evaluate the performance of CoRA, we conduct extensive experiments on several well-established benchmarks. The evaluation covers uni-modal, multi-modal covariate-aware forecasting and multivariate forecasting tasks. The datasets we used are described below:

For uni-modal, long-term, covariate-aware forecasting tasks, we include the following benchmark datasets: ETT (Electricity Transforming Temperature) (Zhou et al., 2021) contains seven power transformer load factors from July 2016 to July 2018. According to sampling frequency and location, the dataset is partitioned into four subsets: ETTh1 and ETTh2 contain hourly measurements, whereas ETTm1 and ETTm2 provide observations at 15-minute intervals. Weather (Wu et al., 2021) comprises 21 meteorological variates collected at 10-minute intervals throughout 2020 from the Max Planck Institute for Biogeochemistry. ECL (Electricity Consuming Load) (Wu et al., 2021) records hourly electricity consumption of 321 residential and commercial clients, offering diverse patterns of consumption behavior. Traffic (Wu et al., 2021) consists of hourly road occupancy data from 862 sensors installed on highways in the San Francisco Bay Area, covering the period January 2015 to December 2016. Further statistics are reported in Table 6.

For uni-modal short-term covariate-aware forecasting task, we include the following benchmark datasets: EPF (Electricity Price Forecasting) (Lago et al., 2021) contains 6 years of hourly day-ahead electricity prices, complemented by two exogenous forecast series (load and renewable generation). The dataset spans five major European electricity markets, facilitating robust cross-market performance analysis under diverse price dynamics and market conditions. (1) NP (Nord Pool) covers the Nord Pool electricity market, containing hourly electricity prices together with grid load and wind power forecasts from 2013-01-01 to 2018-12-24. (2) PJM corresponds to the Pennsylvania–New Jersey–Maryland market, including the zonal electricity price in the Commonwealth Edison (COMED) area, system load, and COMED load forecasts from 2013-01-01 to 2018-12-24. (3) BE denotes Belgium's electricity market, recording hourly electricity prices, load forecasts in Belgium, and generation forecasts in France from 2011-01-09 to 2016-12-31. (4) FR corresponds to the French electricity market, containing hourly prices with associated load and generation forecasts from 2012-01-09 to 2017-12-31. (5) DE represents the German electricity market, providing hourly prices, zonal load forecasts in the TSO Amprion zone, and wind and solar generation forecasts from 2012-01-09 to 2017-12-31. Further statistics are reported in Table 6.

To assess CoRA's capability in multi-modal covariate-aware forecasting, we employ RT-1 (**?**), a large-scale robotic dataset with about 130k demonstrations collected over 17 months using 13 robots in office kitchen environments. It covers 744 skills, ranging from basic object manipulation to long-horizon instructions, each paired with natural language commands and visual observations. The dataset provides rich multi-modal supervision, supporting studies on instruction-conditioned and multi-modal forecasting. The RT-1 dataset is particularly valuable for studying multi-modal and instruction-conditioned forecasting, as it provides paired visual observations and natural language descriptions aligned with robotic trajectories. In our experiments, we use a subset of RT-1, specifically the 'Move Object Near Object' skill, and further restrict it to series with lengths no shorter than 45. Each sequence is partitioned into training, validation, and test sets by assigning the last four points as test targets and the preceding four points as validation targets, with the remaining points used for training. This protocol guarantees at least one validation and one test instance per series, under a setup with an input length of 32 and a prediction horizon of 4. Time-MMD (**?**) is a large-scale multi-modal dataset encompassing nine diverse domains, including agriculture, climate, healthcare, and transportation. Each time series is paired with corresponding textual information sourced from curated domain reports and structured web search results, enabling evaluation of text-enhanced forecasting performance. For consistency with prior work (Han et al., 2025), we exclude the Agriculture and Economy subsets, and keep all other experimental settings identical to the official configuration. Details of these datasets are provided in Table 7.

Table 6: Detailed dataset descriptions. *Nums* denotes the number of covariates. *Freq* denotes the sampling interval of time points. The dataset size is given as (Train, Validation, Test).

| Dataset | Domain | Nums | Freq | Target Variate | Covariate | Dataset Size | Prediction Horizon |
|---------|--------|------|------|----------------|-----------|--------------|--------------------|
| Electricity | Energy | 320 | 1H | Electricity Consumption | Electricity Consumption | (18317, 2633, 5261) | (96, 192, 336, 720) |
| Weather | Weather | 20 | 10M | $CO_2$ Concentration | Climate Feature | (36792, 5271, 10540) | (96, 192, 336, 720) |
| ETTh | Energy | 6 | 1H | Oil Temperature | Power Load Feature | (8545, 2881, 2881) | (96, 192, 336, 720) |
| ETTm | Energy | 6 | 15M | Oil Temperature | Power Load Feature | (34465, 11521, 11521) | (96, 192, 336, 720) |
| Traffic | Traffic | 861 | 1H | Road Occupancy Rates | Road Occupancy Rates | (12185, 1757, 3509) | (96, 192, 336, 720) |
| NP | Electricity | 2 | 1H | Nord Pool Electricity Price | Grid Load, Wind Power | (36500, 5219, 10460) | 24 |
| PJM | Electricity | 2 | 1H | PJM Electricity Price | System Load, Zonal COMED Load | (36500, 5219, 10460) | 24 |
| BE | Electricity | 2 | 1H | Belgium Electricity Price | Generation, System Load | (36500, 5219, 10460) | 24 |
| FR | Electricity | 2 | 1H | France Electricity Price | Generation, System Load | (36500, 5219, 10460) | 24 |
| DE | Electricity | 2 | 1H | German Electricity Price | Wind Power, Amprion Zonal Load | (36500, 5219, 10460) | 24 |

Table 7: Detailed descriptions of RT-1 (Brohan et al., 2022) and TimeMMD (**?**).

| Dataset | Domain | Num. Obs. | Num. Series | Freq | Target Variate | Covariate Type | Prediction Horizon |
|---------|--------|-----------|-------------|------|----------------|----------------|--------------------|
| RT-1 | Solar Power | 33,420 | 2871 | $\frac{1}{3}$S | height to bottom | Image | 4 |
| TimeMMD | Agriculture | 486 | 1 | 1M | Retail Broiler Composite | Text | 12 |
| | Climate | 496 | 1 | 1M | Drought Level | Text | 12 |
| | Economy | 423 | 1 | 1M | International Trade Balance | Text | 12 |
| | Energy | 1479 | 1 | 1M | Gasoline Prices | Text | 12 |
| | Environment | 11102 | 1 | 1M | Air Quality Index | Text | 12 |
| | Health | 1389 | 1 | 1W | Influenza Patients Proportion | Text | 12 |
| | Security | 297 | 1 | 1D | Disaster and Emergency Grants | Text | 12 |
| | Social Good | 900 | 1 | 1M | Unemployment Rate | Text | 12 |
| | Traffic | 531 | 1 | 1M | Travel Volume | Text | 12 |

## A.2 BASELINE MODELS

We compared our method to multiple advanced baselines across various forecasting tasks.

**Time Series Foundation Models** We evaluate CoRA across multiple Time Series Foundation Models, including Sundial (Liu et al., 2025), TimesFM (Das et al., 2023b), Chronos-Bolt (Ansari et al., 2024), and FlowState (Graf et al., 2025). Specifically, on the Time-MMD dataset (**?**), we further include Moirai (Liu et al., 2024c) and TabPFN-TS (Hoo et al., 2025) as baselines.

**Covariate-Aware Deep models** We compare CoRA with diverse advanced supervised deep forecasters. These include Transformer-based architectures such as TimeXer (Wang et al., 2024), iTransformer (Liu et al., 2023), PatchTST (Nie et al., 2022), Crossformer (Zhang & Yan, 2022), Autoformer (Wu et al., 2021), TiDE (Das et al., 2023a), Time-LLM (Jin et al., 2023), TTM (Ekambaram et al., 2024) and TFT (Lim et al., 2021); classical sequence models such as N-BEATS (Oreshkin et al., 2019), NBEATSx (Olivares et al., 2023) and DeepAR (Salinas et al., 2020); and other strong baselines including DLinear (Zeng et al., 2023) and SCINet (Liu et al., 2022).

**Adaptation Method** We evaluate CoRA against other covariate adaptation methods, including UniCA (Han et al., 2025), ChronosX (Arango et al., 2025), and AdaPTS (Benechehab et al., 2025). In addition, to assess the role of covariates explicitly, we also compare with LoRA (Hu et al., 2021) and SFT, which adapts model parameters without leveraging covariate signals.

### A.3 IMPLEMENTATION DETAILS

All experiments are conducted using PyTorch on NVIDIA A100 Tensor Core GPUs. We employ the Adam optimizer, along with the respective loss function of each foundation model, for optimization; unless otherwise specified, the default loss function is mean squared error (MSE).

The training process is limited to a maximum of 50 epochs with early stopping, and patience is set to 3. The learning rate is selected from the set {5e-6, 1e-5, 2e-5}, and the batch size is fixed at 128.

For EPF, we follow the benchmark results reported in (Wang et al., 2024). For Time-MMD (**?**), we use the results reported in (Han et al., 2025), both of which are strictly based on the configurations in original papers. For all other results, we reproduce both the adaptation methods and the deep forecasting models from their official repositories, keeping hyperparameters and training configurations unchanged to ensure a fair evaluation of each base model.

---

**Algorithm 1** CoRA Algorithm

---

**Require:** Past target series $\mathbf{x}_{1:T} = \{x_1, \ldots, x_T\}$; Covariates $\mathbf{C}_{1:\tau} = \{\mathbf{C}_1, \ldots, \mathbf{C}_\tau\}$ (time series, text, image); Prediction horizon $H$

1: $\mathbf{E}_{1:\tau_i}^{m_i} = \text{FM-Backbone}(\mathbf{C}_{1:\tau_i}^{m_i}),\ i = 1, \ldots, N,\ m_i \in \{\text{ts}, \text{txt}, \text{img}\}$
2: $\tilde{\mathbf{E}}^{\text{ts}} = \mathbf{E}_\tau^{\text{ts}}$
3: $\tilde{\mathbf{E}}^{\text{txt}} = \frac{1}{\tau} \sum_{t=1}^\tau \mathbf{E}_t^{\text{txt}},\ \tilde{\mathbf{E}}^{\text{img}} = \frac{1}{\tau} \sum_{t=1}^\tau \mathbf{E}_t^{\text{img}}$
4: $\mathbf{E}_{1:T}^{\text{target}} = \text{TSFM-Backbone}(\mathbf{x}_{1:T})$
5: $\tilde{\mathbf{E}}^{\text{target}} = \mathbf{E}_T^{\text{target}}$
6: $\hat{\mathbf{E}}^{m_i} = \tilde{\mathbf{E}}^{m_i} \mathbf{W}^{m_i} + \mathbf{b}^{m_i},\ i = 1, \ldots, N,\ m_i \in \{\text{ts}, \text{txt}, \text{img}\}$
7: $\hat{\mathbf{E}} = \text{Concat}\left(\hat{\mathbf{E}}^{\text{ts}}, \hat{\mathbf{E}}^{\text{txt}}, \hat{\mathbf{E}}^{\text{img}}\right)$
8: $\mathbf{H} = \text{Softmax}(\mathbf{W}_{\text{CE}}) \cdot \hat{\mathbf{E}}$
9: $\gamma, \beta, \alpha = \text{MLP}\left(\mathbf{H}\right)$
10: $\hat{\mathbf{x}}_{T+1:T+H} = (1 + \alpha)\, \text{TSFM-Head}\left(\gamma + (1 + \beta)\, \tilde{\mathbf{E}}^{\text{target}}\right)$
11: **return** $\hat{\mathbf{x}}_{T+1:T+H}$

---

**Algorithm 2** Granger Causality Algorithm

---

**Require:** covariate series $A$, target series $B$, maximum lag $L_{\max}$, criterion
**Ensure:** Granger causality strength $CE$, selected lag $l$

1: Select lag $l$ by minimizing criterion over $1, \ldots, L_{\max}$
2: Fit restricted model on $B_t$         ▷ use $\{B_{t-1}, \ldots, B_{t-l}\}$, residual variance $\sigma_r^2$
3: Fit unrestricted model on $B_t$   ▷ use $\{B_{t-1}, \ldots, B_{t-l}, A_{t-1}, \ldots, A_{t-l}\}$, residual variance $\sigma_u^2$
4: Compute Granger causality strength: $CE \leftarrow \log \frac{\sigma_r^2}{\sigma_u^2}$
5: **return** CE

---

## B FULL RESULTS

### B.1 FULL RESULTS OF UNI-MODAL COVARIATE-AWARE FORECASTING

Table 8 reports the complete results of the uni-modal covariate-aware forecasting task across widely used datasets. All adaptation methods built on Sundial are fine-tuned only for the output horizon of 720, consistent with the available pre-trained Sundial weights. For shorter horizons, the outputs are obtained by truncating the 720-length predictions. In contrast, the baseline deep models are

individually trained for each prediction length. Overall, adaptation methods on top of TSFMs consistently outperform conventional deep models, and our proposed CoRA achieves state-of-the-art results, demonstrating its effectiveness as a general approach for covariate-aware adaptation.

Table 8: Full results of the long-term covariate-aware forecasting task. For all baselines, the lookback length $L$ is fixed at 2880 and dash (-) denotes out of memory (OOM) problem.

| Models | | CoRA (Ours) | | AdaPTS (2025) | | ChronosX (2025) | | UniCA (2025) | | TimeXer (2024) | | iTransformer (2023) | | PatchTST (2022) | | NBEATSx (2023) | | Crossformer (2023) | | DLinear (2023) | |
|---|---|---|---|---|---|---|---|---|---|---|---|---|---|---|---|---|---|---|---|---|---|
| Metric | | MSE | MAE | MSE | MAE | MSE | MAE | MSE | MAE | MSE | MAE | MSE | MAE | MSE | MAE | MSE | MAE | MSE | MAE | MSE | MAE |
| ETTh1 | 96 | 0.051 | 0.171 | 0.054 | 0.174 | 0.066 | 0.195 | 0.055 | 0.174 | 0.078 | 0.227 | 0.075 | 0.219 | 0.080 | 0.229 | 0.153 | 0.326 | 0.167 | 0.340 | 0.156 | 0.316 |
| | 192 | 0.064 | 0.197 | 0.068 | 0.199 | 0.075 | 0.213 | 0.070 | 0.201 | 0.084 | 0.235 | 0.114 | 0.270 | 0.084 | 0.235 | 0.176 | 0.349 | 0.299 | 0.463 | 0.176 | 0.338 |
| | 336 | 0.071 | 0.210 | 0.079 | 0.220 | 0.086 | 0.232 | 0.085 | 0.227 | 0.090 | 0.244 | 0.160 | 0.324 | 0.088 | 0.239 | 0.206 | 0.383 | 0.500 | 0.565 | 0.211 | 0.378 |
| | 720 | 0.086 | 0.233 | 0.101 | 0.252 | 0.113 | 0.268 | 0.128 | 0.286 | 0.102 | 0.255 | 0.292 | 0.455 | 0.130 | 0.291 | 0.190 | 0.347 | 0.576 | 0.635 | 0.507 | 0.598 |
| | Avg | 0.068 | 0.203 | 0.076 | 0.211 | 0.085 | 0.227 | 0.085 | 0.222 | 0.089 | 0.240 | 0.160 | 0.317 | 0.096 | 0.249 | 0.181 | 0.351 | 0.386 | 0.501 | 0.263 | 0.408 |
| ETTh2 | 96 | 0.111 | 0.258 | 0.112 | 0.256 | 0.258 | 0.389 | 0.125 | 0.272 | 0.168 | 0.329 | 0.175 | 0.339 | 0.188 | 0.349 | 0.245 | 0.407 | 0.270 | 0.410 | 0.250 | 0.402 |
| | 192 | 0.136 | 0.291 | 0.143 | 0.297 | 0.309 | 0.429 | 0.165 | 0.321 | 0.186 | 0.348 | 0.214 | 0.381 | 0.184 | 0.346 | 0.176 | 0.349 | 0.348 | 0.481 | 0.317 | 0.453 |
| | 336 | 0.149 | 0.311 | 0.157 | 0.317 | 0.353 | 0.462 | 0.199 | 0.359 | 0.192 | 0.355 | 0.304 | 0.455 | 0.190 | 0.348 | 0.206 | 0.383 | 0.383 | 0.509 | 0.323 | 0.460 |
| | 720 | 0.169 | 0.335 | 0.213 | 0.372 | 0.538 | 0.585 | 0.298 | 0.447 | 0.231 | 0.386 | 0.536 | 0.606 | 0.203 | 0.366 | 0.190 | 0.347 | 0.578 | 0.607 | 0.388 | 0.500 |
| | Avg | 0.141 | 0.299 | 0.156 | 0.311 | 0.365 | 0.466 | 0.197 | 0.350 | 0.194 | 0.355 | 0.307 | 0.445 | 0.191 | 0.352 | 0.181 | 0.351 | 0.395 | 0.502 | 0.320 | 0.454 |
| ETTm1 | 96 | 0.026 | 0.122 | 0.027 | 0.123 | 0.028 | 0.124 | 0.030 | 0.128 | 0.038 | 0.147 | 0.036 | 0.148 | 0.031 | 0.134 | 0.066 | 0.199 | 0.038 | 0.154 | 0.030 | 0.131 |
| | 192 | 0.039 | 0.149 | 0.041 | 0.156 | 0.044 | 0.157 | 0.045 | 0.158 | 0.062 | 0.194 | 0.053 | 0.178 | 0.049 | 0.172 | 0.076 | 0.226 | 0.055 | 0.185 | 0.052 | 0.176 |
| | 336 | 0.048 | 0.165 | 0.054 | 0.181 | 0.057 | 0.181 | 0.056 | 0.177 | 0.069 | 0.203 | 0.065 | 0.197 | 0.061 | 0.195 | 0.203 | 0.381 | 0.077 | 0.219 | 0.069 | 0.201 |
| | 720 | 0.058 | 0.182 | 0.063 | 0.198 | 0.067 | 0.199 | 0.068 | 0.202 | 0.080 | 0.225 | 0.080 | 0.221 | 0.077 | 0.224 | 0.102 | 0.264 | 0.102 | 0.271 | 0.086 | 0.226 |
| | Avg | 0.043 | 0.155 | 0.046 | 0.165 | 0.049 | 0.165 | 0.050 | 0.166 | 0.062 | 0.192 | 0.059 | 0.186 | 0.055 | 0.181 | 0.112 | 0.268 | 0.068 | 0.207 | 0.059 | 0.184 |
| ETTm2 | 96 | 0.059 | 0.175 | 0.059 | 0.177 | 0.063 | 0.182 | 0.075 | 0.199 | 0.105 | 0.240 | 0.097 | 0.243 | 0.081 | 0.213 | 0.181 | 0.338 | 0.164 | 0.327 | 0.071 | 0.197 |
| | 192 | 0.085 | 0.218 | 0.094 | 0.228 | 0.092 | 0.228 | 0.106 | 0.246 | 0.153 | 0.295 | 0.135 | 0.286 | 0.123 | 0.272 | 0.204 | 0.364 | 0.193 | 0.344 | 0.107 | 0.250 |
| | 336 | 0.108 | 0.251 | 0.121 | 0.265 | 0.117 | 0.263 | 0.133 | 0.282 | 0.195 | 0.340 | 0.164 | 0.322 | 0.146 | 0.297 | 0.242 | 0.408 | 0.194 | 0.364 | 0.135 | 0.285 |
| | 720 | 0.146 | 0.302 | 0.155 | 0.310 | 0.153 | 0.310 | 0.173 | 0.332 | 0.191 | 0.340 | 0.198 | 0.363 | 0.173 | 0.328 | 0.259 | 0.424 | 0.281 | 0.429 | 0.177 | 0.332 |
| | Avg | 0.100 | 0.237 | 0.107 | 0.245 | 0.106 | 0.246 | 0.122 | 0.265 | 0.161 | 0.304 | 0.149 | 0.304 | 0.131 | 0.278 | 0.222 | 0.384 | 0.208 | 0.366 | 0.123 | 0.266 |
| Weather | 96 | 0.001 | 0.020 | 0.001 | 0.021 | 0.001 | 0.028 | 0.001 | 0.027 | 0.002 | 0.031 | 0.002 | 0.033 | 0.002 | 0.034 | 0.008 | 0.076 | 0.003 | 0.040 | 0.007 | 0.072 |
| | 192 | 0.001 | 0.025 | 0.001 | 0.025 | 0.002 | 0.031 | 0.002 | 0.031 | 0.002 | 0.032 | 0.002 | 0.033 | 0.002 | 0.035 | 0.105 | 0.092 | 0.003 | 0.042 | 0.008 | 0.076 |
| | 336 | 0.002 | 0.028 | 0.002 | 0.028 | 0.002 | 0.034 | 0.002 | 0.034 | 0.002 | 0.033 | 0.002 | 0.034 | 0.002 | 0.034 | 0.009 | 0.085 | 0.004 | 0.051 | 0.008 | 0.079 |
| | 720 | 0.002 | 0.032 | 0.002 | 0.032 | 0.002 | 0.037 | 0.002 | 0.039 | 0.002 | 0.034 | 0.002 | 0.036 | 0.003 | 0.041 | 0.010 | 0.090 | 0.005 | 0.056 | 0.008 | 0.078 |
| | Avg | 0.001 | 0.026 | 0.002 | 0.027 | 0.002 | 0.033 | 0.002 | 0.033 | 0.002 | 0.033 | 0.002 | 0.034 | 0.002 | 0.036 | 0.033 | 0.086 | 0.004 | 0.047 | 0.008 | 0.076 |
| ECL | 96 | 0.159 | 0.279 | 0.173 | 0.295 | 0.176 | 0.289 | 0.175 | 0.296 | 0.235 | 0.339 | 0.248 | 0.368 | 0.248 | 0.370 | 0.306 | 0.415 | 0.285 | 0.388 | 0.222 | 0.341 |
| | 192 | 0.187 | 0.305 | 0.199 | 0.315 | 0.202 | 0.315 | 0.207 | 0.321 | 0.271 | 0.364 | 0.291 | 0.403 | 0.332 | 0.437 | 0.338 | 0.439 | 0.382 | 0.475 | 0.255 | 0.364 |
| | 336 | 0.208 | 0.325 | 0.226 | 0.340 | 0.215 | 0.331 | 0.240 | 0.352 | 0.309 | 0.398 | 0.316 | 0.425 | 0.365 | 0.458 | 0.373 | 0.463 | 0.362 | 0.447 | 0.287 | 0.393 |
| | 720 | 0.223 | 0.345 | 0.249 | 0.367 | 0.232 | 0.355 | 0.298 | 0.419 | 0.354 | 0.446 | 0.318 | 0.426 | 0.362 | 0.460 | 0.390 | 0.478 | 0.377 | 0.474 | 0.292 | 0.404 |
| | Avg | 0.194 | 0.314 | 0.212 | 0.329 | 0.206 | 0.323 | 0.230 | 0.347 | 0.292 | 0.387 | 0.293 | 0.406 | 0.327 | 0.431 | 0.352 | 0.449 | 0.352 | 0.446 | 0.264 | 0.376 |
| Traffic | 96 | 0.101 | 0.169 | - | - | - | - | 0.109 | 0.185 | 0.149 | 0.250 | 0.124 | 0.210 | 0.146 | 0.245 | 0.187 | 0.291 | 0.164 | 0.259 | 0.164 | 0.270 |
| | 192 | 0.109 | 0.179 | - | - | - | - | 0.118 | 0.197 | 0.156 | 0.258 | 0.131 | 0.221 | 0.152 | 0.253 | 0.210 | 0.316 | 0.225 | 0.317 | 0.179 | 0.290 |
| | 336 | 0.111 | 0.187 | - | - | - | - | 0.121 | 0.204 | 0.154 | 0.258 | 0.136 | 0.232 | 0.152 | 0.255 | 0.224 | 0.333 | 0.297 | 0.375 | 0.190 | 0.308 |
| | 720 | 0.128 | 0.208 | - | - | - | - | 0.141 | 0.226 | 0.168 | 0.271 | 0.163 | 0.265 | 0.165 | 0.267 | 0.267 | 0.372 | 0.411 | 0.378 | 0.280 | 0.400 |
| | Avg | 0.112 | 0.186 | - | - | - | - | 0.122 | 0.203 | 0.157 | 0.259 | 0.139 | 0.232 | 0.154 | 0.255 | 0.222 | 0.328 | 0.274 | 0.332 | 0.203 | 0.317 |

## B.2 FULL RESULTS OF MULTI-MODAL COVARIATE-AWARE FORECASTING

Table 9 reports the full results on the Time-MMD benchmark. We employ the Qwen3-Embedding (Zhang et al., 2025) as a backbone in CoRA to derive text embeddings. Compared to Sundial (Liu et al., 2025) in the zero-shot setting and Unica (Han et al., 2025), CoRA consistently achieves superior performance across both deterministic metrics (MSE, MAE) and probabilistic metrics (CRPS). This demonstrates that CoRA successfully captures meaningful interactions between temporal dynamics and textual covariates. These results further highlight the strength of CoRA as a general and powerful strategy for integrating multi-modal information into TSFMs.

Table 9: Full results of multi-modal covariate-aware forecasting task on TimeMMD dataset.

| | Models | CoRA (Ours) | UniCA (2025) | Sundial (2025) | NBEATS (2023) | PatchTST (2022) | DeepAR (2020) | TFT (2021) | TiDE (2023a) | Time-LLM (2023) | TTM (2024) | Moirai (2024) | TabPFN-TS (2025) |
|---|---|---|---|---|---|---|---|---|---|---|---|---|---|
| Average | Average | **0.641** | 0.661 | 0.662 | 0.882 | 0.933 | 1.361 | 0.947 | 0.927 | 0.835 | 0.820 | 0.751 | 0.795 |
| | MSE | **0.580** | 0.591 | 0.591 | 0.782 | 0.793 | 1.605 | 0.992 | 0.869 | 0.723 | 0.685 | 0.696 | 0.787 |
| | MAE | **0.690** | 0.716 | 0.716 | 0.884 | 1.009 | 1.219 | 0.958 | 0.976 | 0.847 | 0.866 | 0.821 | 0.837 |
| | CRPS | **0.653** | 0.677 | 0.678 | 0.980 | 0.996 | 1.260 | 0.891 | 0.937 | 0.935 | 0.909 | 0.735 | 0.762 |
| Climate | Average | 0.536 | 0.567 | 0.567 | 0.668 | 0.724 | 0.737 | 0.695 | 0.575 | 0.634 | 0.526 | 0.596 | **0.525** |
| | MSE | 0.440 | 0.487 | 0.487 | 0.519 | 0.640 | 0.623 | 0.599 | 0.465 | 0.468 | 0.408 | 0.488 | **0.407** |
| | MAE | **0.562** | 0.595 | 0.595 | 0.712 | 0.788 | 0.779 | 0.768 | 0.685 | 0.687 | 0.635 | 0.706 | 0.638 |
| | CRPS | 0.607 | 0.620 | 0.620 | 0.773 | 0.743 | 0.809 | 0.719 | 0.574 | 0.746 | 0.535 | 0.593 | **0.529** |
| Energy | Average | **0.888** | 0.892 | 0.892 | 1.611 | 1.274 | 3.768 | 1.018 | 1.303 | 1.253 | 1.216 | 1.011 | 1.233 |
| | MSE | **0.838** | 0.846 | 0.846 | 1.706 | 1.305 | 6.328 | 1.047 | 1.391 | 1.217 | 1.019 | 1.024 | 1.370 |
| | MAE | **0.928** | 0.930 | 0.930 | 1.429 | 1.252 | 2.368 | 1.004 | 1.138 | 1.161 | 1.042 | 1.035 | 1.163 |
| | CRPS | **0.897** | 0.900 | 0.900 | 1.699 | 1.266 | 2.607 | 1.004 | 1.379 | 1.380 | 1.587 | 0.975 | 1.167 |
| Environment | Average | **0.604** | 0.608 | 0.608 | 0.725 | 0.644 | 0.689 | 0.638 | 0.638 | 0.699 | 0.644 | 0.641 | 0.644 |
| | MSE | 0.527 | **0.519** | **0.519** | 0.628 | 0.589 | 0.648 | 0.601 | 0.572 | 0.617 | 0.546 | 0.623 | 0.611 |
| | MAE | **0.730** | 0.742 | 0.742 | 0.809 | 0.785 | 0.822 | 0.763 | 0.778 | 0.774 | 0.777 | 0.756 | 0.772 |
| | CRPS | 0.554 | 0.564 | 0.564 | 0.739 | 0.558 | 0.596 | 0.550 | 0.564 | 0.707 | 0.609 | **0.543** | 0.550 |
| Health | Average | **0.609** | 0.637 | 0.637 | 0.873 | 0.930 | 1.131 | 1.014 | 0.973 | 0.862 | 0.966 | 0.776 | 0.969 |
| | MSE | **0.487** | 0.514 | 0.513 | 0.739 | 0.874 | 1.023 | 1.059 | 0.916 | 0.735 | 0.906 | 0.722 | 0.964 |
| | MAE | **0.687** | 0.706 | 0.706 | 0.860 | 0.928 | 1.118 | 1.004 | 0.992 | 0.846 | 0.989 | 0.821 | 1.008 |
| | CRPS | **0.653** | 0.692 | 0.692 | 1.020 | 0.989 | 1.251 | 0.979 | 1.010 | 1.004 | 1.002 | 0.786 | 0.936 |
| Security | Average | **0.657** | 0.688 | 0.689 | 0.847 | 1.170 | 1.419 | 1.399 | 1.521 | 0.862 | 0.763 | 0.746 | 0.678 |
| | MSE | **0.595** | 0.620 | 0.620 | 0.692 | 0.882 | 1.078 | 1.614 | 1.260 | 0.690 | 0.676 | 0.669 | 0.612 |
| | MAE | **0.736** | 0.763 | 0.764 | 0.927 | 1.332 | 1.607 | 1.409 | 1.767 | 0.951 | 0.880 | 0.856 | 0.764 |
| | CRPS | **0.641** | 0.682 | 0.683 | 0.922 | 1.295 | 1.571 | 1.175 | 1.535 | 0.946 | 0.732 | 0.714 | 0.657 |
| SocialGood | Average | **0.745** | 0.778 | 0.778 | 0.863 | 1.219 | 1.386 | 1.264 | 0.952 | 1.052 | 0.980 | 0.781 | 0.903 |
| | MSE | 0.784 | 0.762 | 0.762 | 0.780 | 0.877 | 1.231 | 1.469 | 0.973 | 0.932 | 0.816 | **0.735** | 0.917 |
| | MAE | **0.719** | 0.788 | 0.788 | 0.843 | 1.347 | 1.403 | 1.172 | 0.943 | 1.036 | 1.062 | 0.803 | 0.912 |
| | CRPS | **0.733** | 0.784 | 0.785 | 0.967 | 1.434 | 1.523 | 1.150 | 0.941 | 1.188 | 1.061 | 0.804 | 0.881 |
| Traffic | Average | 0.448 | 0.458 | 0.458 | 0.584 | 0.569 | **0.401** | 0.599 | 0.529 | 0.484 | 0.647 | 0.704 | 0.616 |
| | MSE | 0.390 | 0.387 | 0.387 | 0.408 | 0.385 | **0.305** | 0.552 | 0.506 | 0.401 | 0.428 | 0.610 | 0.631 |
| | MAE | 0.470 | 0.488 | 0.488 | 0.608 | 0.632 | **0.435** | 0.589 | 0.528 | 0.475 | 0.679 | 0.772 | 0.605 |
| | CRPS | 0.484 | 0.498 | 0.498 | 0.737 | 0.689 | **0.462** | 0.657 | 0.553 | 0.576 | 0.834 | 0.731 | 0.611 |

## B.3 FULL RESULTS OF MULTIVARIATE FORECASTING

Table 10 summarizes results of multivariate forecasting across seven widely used datasets. On this benchmark, CoRA achieves state-of-the-art performance across all datasets, substantially improving upon recent deep forecasters. These results demonstrate that CoRA can jointly predict multiple target variables in a unified manner, highlighting its effectiveness as a general adaptation strategy.

## B.4 FULL RESULTS OF GENERALITY

We conduct extensive experiments on the EPF dataset using several representative TSFMs. As shown in Table 11, CoRA consistently improves the performance of all TSFMs across both MSE and MAE metrics. Compared with their zero-shot baselines, the improvements are significant, demonstrating the generality and effectiveness of CoRA as a universal covariate adaptation method. We report results under the same training configuration and additionally provide the relative improvement ratio in MSE as a more intuitive assessment of the benefits brought by CoRA.

Table 10: Full results of the multivariate forecasting task. For all baselines, the look-back length $L$ is fixed at 2880, and $Avg$ means the average results from all four prediction lengths.

| Models | | CoRA (Ours) | | Timer-XL (2024d) | | TimeXer (2024) | | iTransformer (2023) | | PatchTST (2022) | | Crossformer (2023) | | TiDE (2023a) | | DLinear (2023) | | SCINet (2022) | | Autoformer (2021) | |
|---|---|---|---|---|---|---|---|---|---|---|---|---|---|---|---|---|---|---|---|---|---|
| Metric | | MSE | MAE | MSE | MAE | MSE | MAE | MSE | MAE | MSE | MAE | MSE | MAE | MSE | MAE | MSE | MAE | MSE | MAE | MSE | MAE |
| ETTh1 | 96 | 0.344 | 0.381 | 0.483 | 0.485 | 0.411 | 0.438 | 0.436 | 0.466 | 0.428 | 0.450 | 0.479 | 0.494 | 0.565 | 0.536 | 0.433 | 0.451 | 0.713 | 0.625 | 0.630 | 0.524 |
| | 192 | 0.387 | 0.408 | 0.520 | 0.506 | 0.442 | 0.459 | 0.469 | 0.487 | 0.476 | 0.477 | 0.587 | 0.550 | 0.634 | 0.572 | 0.479 | 0.482 | 0.736 | 0.638 | 0.762 | 0.519 |
| | 336 | 0.412 | 0.425 | 0.540 | 0.564 | 0.477 | 0.484 | 0.510 | 0.515 | 0.519 | 0.504 | 0.641 | 0.600 | 0.672 | 0.593 | 0.533 | 0.519 | 0.773 | 0.658 | 0.886 | 0.766 |
| | 720 | 0.471 | 0.473 | 0.647 | 0.633 | 0.639 | 0.572 | 0.619 | 0.591 | 0.639 | 0.586 | 0.867 | 0.732 | 0.751 | 0.646 | 0.633 | 0.596 | 0.897 | 0.717 | 0.971 | 0.836 |
| | Avg | 0.404 | 0.422 | 0.548 | 0.547 | 0.492 | 0.488 | 0.508 | 0.515 | 0.516 | 0.504 | 0.643 | 0.594 | 0.656 | 0.587 | 0.519 | 0.512 | 0.780 | 0.660 | 0.812 | 0.661 |
| ETTh2 | 96 | 0.271 | 0.329 | 0.314 | 0.378 | 0.350 | 0.409 | 0.344 | 0.414 | 0.369 | 0.426 | 0.725 | 0.622 | 0.442 | 0.468 | 0.458 | 0.474 | 0.544 | 0.535 | 0.731 | 0.635 |
| | 192 | 0.328 | 0.373 | 0.387 | 0.428 | 0.414 | 0.452 | 0.408 | 0.454 | 0.469 | 0.491 | 0.771 | 0.684 | 0.509 | 0.504 | 0.547 | 0.521 | 0.614 | 0.570 | 0.789 | 0.677 |
| | 336 | 0.353 | 0.397 | 0.445 | 0.473 | 0.455 | 0.482 | 0.473 | 0.502 | 0.563 | 0.548 | 0.852 | 0.701 | 0.582 | 0.549 | 0.667 | 0.619 | 0.669 | 0.596 | 0.898 | 0.738 |
| | 720 | 0.372 | 0.424 | 0.541 | 0.538 | 0.595 | 0.561 | 0.533 | 0.533 | 0.558 | 0.547 | 0.893 | 0.755 | 0.688 | 0.606 | 0.808 | 0.743 | 0.841 | 0.667 | 0.941 | 0.778 |
| | Avg | 0.331 | 0.381 | 0.422 | 0.454 | 0.454 | 0.476 | 0.440 | 0.476 | 0.490 | 0.503 | 0.810 | 0.691 | 0.555 | 0.532 | 0.620 | 0.589 | 0.667 | 0.592 | 0.840 | 0.707 |
| ETTm1 | 96 | 0.294 | 0.336 | 0.313 | 0.374 | 0.356 | 0.397 | 0.342 | 0.388 | 0.333 | 0.382 | 0.330 | 0.384 | 0.317 | 0.364 | 0.311 | 0.358 | 0.391 | 0.427 | 0.740 | 0.627 |
| | 192 | 0.325 | 0.361 | 0.358 | 0.403 | 0.388 | 0.416 | 0.363 | 0.402 | 0.375 | 0.408 | 0.363 | 0.403 | 0.352 | 0.387 | 0.341 | 0.377 | 0.410 | 0.438 | 0.858 | 0.696 |
| | 336 | 0.347 | 0.380 | 0.397 | 0.430 | 0.403 | 0.429 | 0.386 | 0.416 | 0.442 | 0.453 | 0.413 | 0.445 | 0.371 | 0.397 | 0.366 | 0.392 | 0.431 | 0.450 | 0.895 | 0.705 |
| | 720 | 0.381 | 0.407 | 0.456 | 0.469 | 0.444 | 0.455 | 0.423 | 0.444 | 0.449 | 0.453 | 0.639 | 0.596 | 0.413 | 0.422 | 0.410 | 0.422 | 0.468 | 0.472 | 0.934 | 0.700 |
| | Avg | 0.337 | 0.371 | 0.381 | 0.419 | 0.398 | 0.424 | 0.379 | 0.413 | 0.400 | 0.424 | 0.436 | 0.457 | 0.363 | 0.393 | 0.357 | 0.387 | 0.425 | 0.447 | 0.857 | 0.682 |
| ETTm2 | 96 | 0.167 | 0.252 | 0.225 | 0.319 | 0.189 | 0.287 | 0.189 | 0.285 | 0.182 | 0.280 | 0.391 | 0.469 | 0.198 | 0.294 | 0.165 | 0.262 | 0.242 | 0.337 | 0.381 | 0.453 |
| | 192 | 0.224 | 0.295 | 0.291 | 0.366 | 0.249 | 0.330 | 0.238 | 0.318 | 0.238 | 0.317 | 0.475 | 0.514 | 0.323 | 0.387 | 0.220 | 0.304 | 0.282 | 0.361 | 0.449 | 0.493 |
| | 336 | 0.278 | 0.334 | 0.344 | 0.402 | 0.291 | 0.352 | 0.298 | 0.356 | 0.311 | 0.368 | 0.663 | 0.674 | 0.332 | 0.390 | 0.268 | 0.338 | 0.322 | 0.387 | 0.503 | 0.521 |
| | 720 | 0.354 | 0.388 | 0.412 | 0.445 | 0.368 | 0.402 | 0.377 | 0.407 | 0.437 | 0.454 | 0.745 | 0.716 | 0.372 | 0.409 | 0.410 | 0.435 | 0.385 | 0.426 | 0.494 | 0.514 |
| | Avg | 0.256 | 0.317 | 0.318 | 0.383 | 0.274 | 0.343 | 0.276 | 0.342 | 0.292 | 0.355 | 0.569 | 0.593 | 0.306 | 0.370 | 0.266 | 0.335 | 0.308 | 0.378 | 0.457 | 0.495 |
| Weather | 96 | 0.158 | 0.206 | 0.255 | 0.299 | 0.186 | 0.246 | 0.187 | 0.252 | 0.160 | 0.219 | 0.159 | 0.212 | 0.171 | 0.231 | 0.169 | 0.230 | 0.168 | 0.231 | 0.400 | 0.433 |
| | 192 | 0.201 | 0.248 | 0.315 | 0.344 | 0.233 | 0.286 | 0.231 | 0.291 | 0.210 | 0.265 | 0.198 | 0.263 | 0.211 | 0.265 | 0.210 | 0.267 | 0.216 | 0.274 | 0.447 | 0.448 |
| | 336 | 0.249 | 0.288 | 0.331 | 0.366 | 0.281 | 0.318 | 0.273 | 0.325 | 0.273 | 0.309 | 0.246 | 0.298 | 0.253 | 0.296 | 0.257 | 0.310 | 0.299 | 0.333 | 0.462 | 0.452 |
| | 720 | 0.311 | 0.333 | 0.361 | 0.384 | 0.347 | 0.361 | 0.314 | 0.352 | 0.359 | 0.368 | 0.335 | 0.369 | 0.300 | 0.332 | 0.314 | 0.357 | 0.314 | 0.344 | 0.693 | 0.616 |
| | Avg | 0.230 | 0.269 | 0.316 | 0.348 | 0.262 | 0.303 | 0.251 | 0.305 | 0.251 | 0.290 | 0.235 | 0.285 | 0.234 | 0.281 | 0.237 | 0.291 | 0.249 | 0.296 | 0.500 | 0.487 |
| ECL | 96 | 0.124 | 0.220 | 0.131 | 0.229 | 0.137 | 0.241 | 0.167 | 0.275 | 0.136 | 0.240 | 0.133 | 0.232 | 0.130 | 0.226 | 0.129 | 0.227 | 0.144 | 0.252 | 0.256 | 0.362 |
| | 192 | 0.142 | 0.238 | 0.147 | 0.244 | 0.154 | 0.256 | 0.177 | 0.283 | 0.151 | 0.254 | 0.162 | 0.266 | 0.146 | 0.242 | 0.144 | 0.242 | 0.163 | 0.271 | 0.267 | 0.371 |
| | 336 | 0.159 | 0.256 | 0.159 | 0.257 | 0.189 | 0.291 | 0.196 | 0.302 | 0.167 | 0.269 | 0.191 | 0.286 | 0.163 | 0.259 | 0.159 | 0.260 | 0.178 | 0.286 | 0.278 | 0.376 |
| | 720 | 0.194 | 0.287 | 0.183 | 0.279 | 0.210 | 0.311 | 0.234 | 0.335 | 0.199 | 0.297 | 0.249 | 0.338 | 0.199 | 0.290 | 0.192 | 0.292 | 0.239 | 0.331 | 0.367 | 0.451 |
| | Avg | 0.155 | 0.250 | 0.155 | 0.252 | 0.172 | 0.275 | 0.194 | 0.299 | 0.163 | 0.265 | 0.184 | 0.281 | 0.160 | 0.254 | 0.156 | 0.255 | 0.181 | 0.285 | 0.292 | 0.390 |
| Traffic | 96 | 0.350 | 0.245 | 0.569 | 0.428 | 0.377 | 0.269 | 0.375 | 0.275 | 0.397 | 0.286 | 0.481 | 0.256 | 0.377 | 0.264 | 0.379 | 0.270 | 0.455 | 0.342 | 0.538 | 0.405 |
| | 192 | 0.372 | 0.257 | 0.570 | 0.513 | 0.387 | 0.274 | 0.395 | 0.284 | 0.410 | 0.293 | 0.492 | 0.270 | 0.390 | 0.269 | 0.392 | 0.276 | 0.462 | 0.346 | 0.776 | 0.468 |
| | 336 | 0.389 | 0.267 | 0.589 | 0.521 | 0.400 | 0.281 | 0.410 | 0.292 | 0.423 | 0.299 | 0.514 | 0.277 | 0.403 | 0.275 | 0.407 | 0.284 | 0.481 | 0.356 | 0.769 | 0.460 |
| | 720 | 0.426 | 0.288 | 0.658 | 0.577 | 0.440 | 0.298 | 0.447 | 0.312 | 0.457 | 0.313 | 0.601 | 0.337 | 0.438 | 0.294 | 0.447 | 0.307 | 0.512 | 0.363 | 0.885 | 0.524 |
| | Avg | 0.384 | 0.265 | 0.597 | 0.510 | 0.401 | 0.281 | 0.407 | 0.291 | 0.422 | 0.298 | 0.522 | 0.285 | 0.402 | 0.276 | 0.406 | 0.284 | 0.478 | 0.352 | 0.742 | 0.464 |

Table 11: Full results of CoRA generalize to other Time Series Foundation Models.

| Datasets | NP | | PJM | | BE | | FR | | DE | | Avg | |
|---|---|---|---|---|---|---|---|---|---|---|---|---|
| Models | MSE | MAE | MSE | MAE | MSE | MAE | MSE | MAE | MSE | MAE | MSE | MAE |
| Sundial | 0.263 | 0.288 | 0.089 | 0.186 | 0.364 | 0.271 | 0.361 | 0.217 | 0.543 | 0.462 | 0.324 | 0.285 |
| + CoRA | 0.222 | 0.246 | 0.073 | 0.165 | 0.339 | 0.236 | 0.357 | 0.206 | 0.401 | 0.388 | 0.278 | 0.248 |
| Promotion | 15.59% | | 17.98% | | 6.87% | | 1.11% | | 26.15% | | 14.20% | |
| TimesFM | 0.255 | 0.271 | 0.085 | 0.182 | 0.383 | 0.252 | 0.398 | 0.206 | 0.526 | 0.456 | 0.329 | 0.273 |
| + CoRA | 0.246 | 0.271 | 0.083 | 0.182 | 0.380 | 0.251 | 0.394 | 0.205 | 0.487 | 0.433 | 0.318 | 0.268 |
| Promotion | 3.53% | | 2.35% | | 0.78% | | 1.01% | | 7.41% | | 3.34% | |
| Chronos-Bolt | 0.246 | 0.265 | 0.082 | 0.178 | 0.356 | 0.239 | 0.357 | 0.191 | 0.494 | 0.442 | 0.307 | 0.263 |
| + CoRA | 0.235 | 0.255 | 0.076 | 0.170 | 0.353 | 0.233 | 0.352 | 0.184 | 0.445 | 0.414 | 0.292 | 0.251 |
| Promotion | 4.47% | | 7.32% | | 0.84% | | 1.40% | | 9.92% | | 4.89% | |
| FlowState | 0.229 | 0.256 | 0.081 | 0.177 | 0.362 | 0.252 | 0.365 | 0.203 | 0.497 | 0.446 | 0.307 | 0.267 |
| + CoRA | 0.225 | 0.253 | 0.078 | 0.177 | 0.355 | 0.243 | 0.364 | 0.199 | 0.464 | 0.424 | 0.297 | 0.259 |
| Promotion | 1.75% | | 3.70% | | 1.93% | | 0.27% | | 6.64% | | 3.26% | |

## C  SHOWCASES

To facilitate a clear comparison among various models, we present additional prediction showcases for uni-modal covariate-aware forecasting in Figure 8. These examples are provided by the following methods: AdaPTS (Benechehab et al., 2025), TimeXer (Wang et al., 2024), and PatchTST (Nie et al., 2022). Of all the models, CoRA delivers the most accurate future series predictions. Additionally, we provide the showcases of multi-modal covariate-aware forecasting in Figure 9.

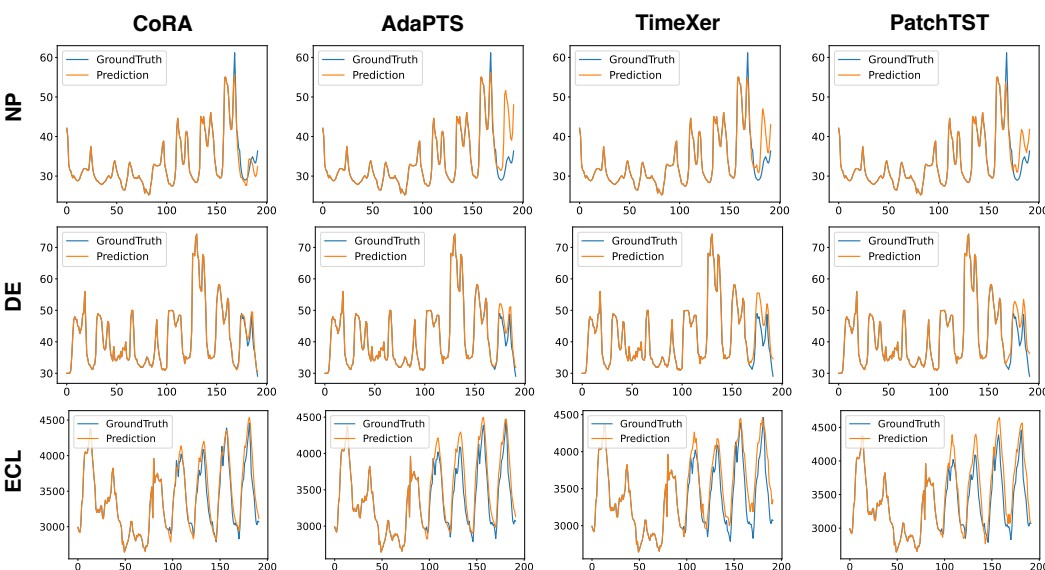

Figure 8: Visualization of uni-modal covariate-aware results on NP, DE and ECL dataset.

## D  LIMITATIONS

A notable limitation of CoRA lies in its treatment of temporally aligned auxiliary modalities such as language and image sequences. At present, CoRA applies a simple mean aggregation along the temporal dimension, which inevitably discards fine-grained temporal dynamics and leads to underutilization of the rich and potentially complementary information contained in these modalities. Future work could investigate more sophisticated fusion strategies that explicitly preserve temporal dependencies, thereby enabling CoRA to more effectively leverage auxiliary modalities and further improve its adaptability across diverse forecasting scenarios.

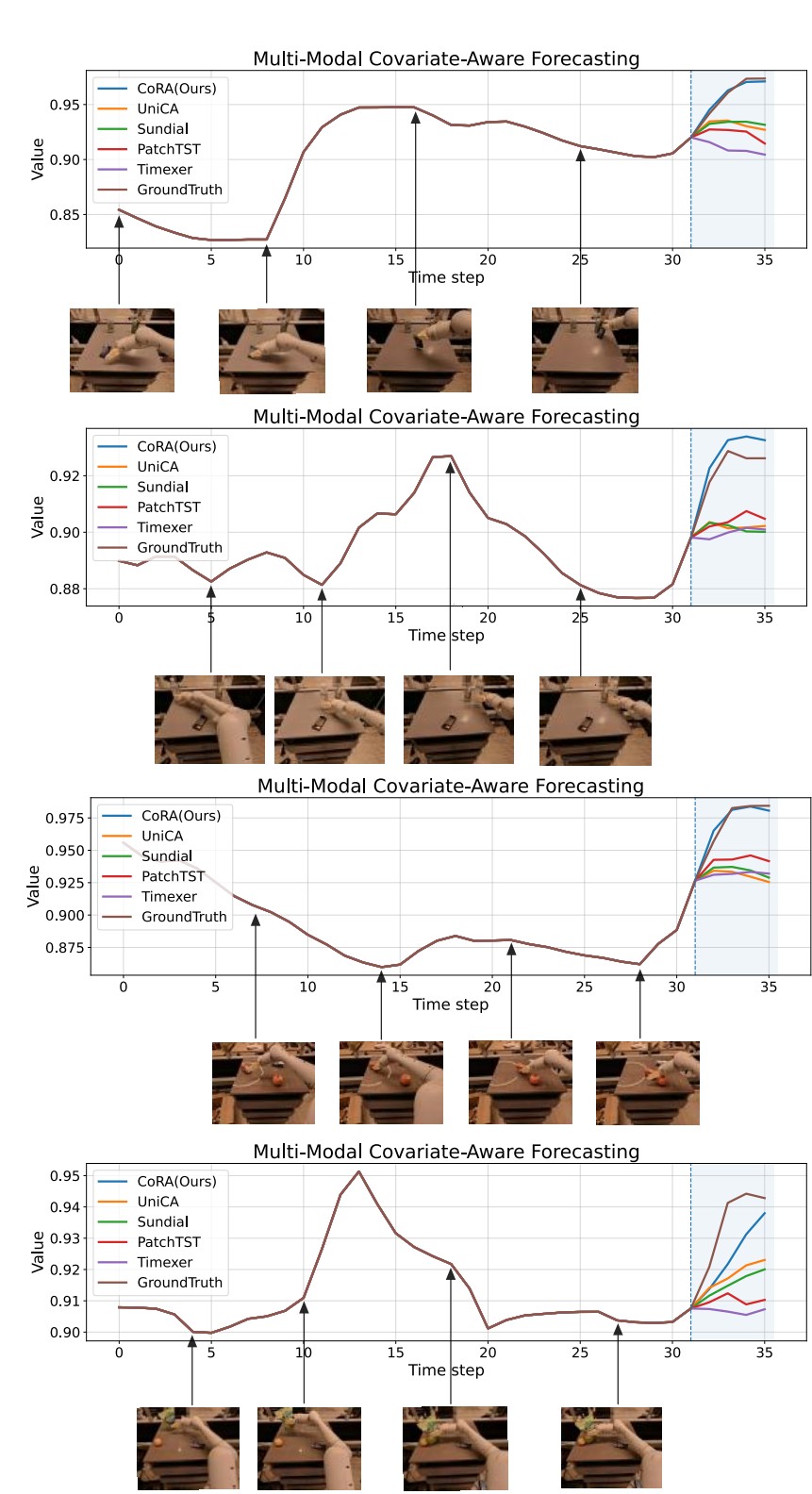

Figure 9: Visualization of multi-modal covariate-aware results on RT-1 dataset.

