# OpenReview forum: "CoRA: Covariate-Aware Adaptation of Time Series Foundation Models"
_ICLR.cc/2026/Conference — Submitted to ICLR 2026_

### Official Review · Reviewer_yv98 · 2025-10-28

**Soundness:** 2
**Presentation:** 2
**Contribution:** 2
**Rating:** 2
**Confidence:** 4

**Summary:**

This paper introduces CoRA, a Covariate-awaRe Adaptation framework for adapting Time Series Foundation Models (TSFMs) to covariate-aware forecasting tasks. CoRA aims to extend the applicability of TSFMs by integrating exogenous covariates from multiple modalities such as time series, text, and images, without modifying the pre-trained backbone. Experiments on a suite of uni-modal and multi-modal benchmarks, including few-shot and multivariate settings, show CoRA consistently outperforming strong supervised and adaptation baselines.

**Strengths:**

1. **Clear Motivation:** The research motivation of this work is clear. The challenge addressed in this study represents a widely recognized concern in the time series domain, bearing significant practical implications.

2. **General Adaptation Mechanism**: CoRA presents a versatile adaptation framework applicable across various TSFMs and covariate modalities, without requiring modifications to the pre-trained model backbones.

3. **Comprehensive Empirical Evaluation:** The experimental section is thorough, including results on uni-modal, multi-modal, and multivariate forecasting tasks. Ablation studies are performed to assess the contribution of each component.

**Weaknesses:**

1. This paper lacks technical novelty and theoretical contributions, as the methods employed are merely simple applications of existing techniques, such as zero-initialization and adaptive layer-normalization (adaLN).

2. The paper does not provide sufficient validation that the discovered "causal relationships" are meaningful or correct. The manuscript lacks a theoretical analysis to interpret the learned relationships and fails to compare its performance against established causal discovery methods. The embeddings could simply be learning spurious correlations.

3. The experimental setup assumes the last dimension is always the endogenous variable. This assumption may be problematic, since the relationship between endogenous and exogenous variables is not necessarily causal and could merely reflect correlations.

4. There is no direct evidence to suggest that the significant performance improvement of CoRA on multi-modal datasets is attributable to its architectural design, rather than the inherent gains from the ViT and Qwen-Embedding components themselves.

**Questions:**

1. This work simply utilizes learnable embeddings to represent weights, a method that captures correlations rather than establishing true causal relationships. The paper lacks a discussion of confounders, instrumental variables, or other causal identification strategies necessary for genuine causal inference. Does a reliable theoretical foundation support this method?

2. The framework is presented as “universally” adaptable, but all adaptation methods in the primary experiments are built on the Sundial TSFM backbone. While some transfer analyses show CoRA applied to other foundation models, direct side-by-side comparisons (with ablations) on all architectures are not exhaustive.  Could the framework's performance depend on the specific architecture of the Sundial model for most tasks ?

3. This paper does not provide a hyperparameter sensitivity analysis or an open-source repository, which means researchers may need to spend a significant amount of extra time adjusting settings to reproduce the results.

---

> ### Author Response · Authors · 2025-11-21
> **Rebuttal of W1~W3**
>
> Many thanks to Reviewer yv98 for providing a valuable review, which helped us improve the quality of our submission.
>
> **W1**: Technical novelty and theoretical contributions.
>
> Thanks for your valuable feedback regarding the novelty. While *Zero-Initialization* and *adaLN* are established techniques, we respectfully argue that our contribution lies in their **principled adaptation to solve unique TSFM challenges**, rather than the components themselves. We clarify our novelty from three aspects:
>
> **1. Rethinking Overlooked Principles in TSFM Adaptation**
>
> Existing attempts to adapt TSFMs with covariates [1,2,3] typically introduce covariate-aware modules that disrupt the original embedding space and do not enforce zero-initialization. As a result, the adapted model no longer starts from behavior consistent with the pre-trained TSFM, often causing unstable training or even performance degradation. This indicates that **principled strategies such as parameter consistency and zero-initialization have not been properly considered in prior TSFM adaptation methods**, which limits their performance.
>
> In contrast, our design **revisits these overlooked principles** and incorporates them explicitly, preserving the pre-trained embedding space and enabling stable and faithful adaptation to covariates.
>
> **2. High-Dimensional Covariate Injection**
>
> Unlike adaptation in language models, covariates in time series are typically multivariate. Effective adaptation therefore requires not only injecting covariates but first **selecting** the relevant ones. Our Granger-causality inspired covariate selection module provides a novel and principled way to handle high-dimensional covariate injection.
>
> **3. Simple yet Effective**
>
> Our method is guided by clear motivations. The resulting modules remain intentionally simple, because simplicity is a feature, not a limitation. We aim to provide a method that is **easy to use, principled in design, and delivers strong performance** without relying on unnecessarily complex architectures.
>
> [1] :  AdaPTS: Adapting Univariate Foundation Models to Probabilistic Multivariate Time Series Forecasting.
>
> [2] : ChronosX: Adapting Pretrained Time Series Models with Exogenous Variables.
>
> [3] : UniCA: Adapting Time Series Foundation Model to General Covariate-Aware Forecasting.
>
> **W2**: Granger Causality relationship.
>
> We appreciate your questions regarding the learned Granger causality relationships. However, there is an important clarification we would like to make to avoid a potential misunderstanding: **Granger causality does not represent causal relationships in the traditional sense** (i.e., “C causes changes in x”). Instead, it asks **whether using a covariate $C$ together with $x_{1:T}$ improves the prediction of $x_{T+1:T+H}$ compared to using $x_{1:T}$ alone**. If it does, then $C$ is said to *Granger-cause* $x$.
>
> Therefore, the learned Granger causality reflects **the predictive usefulness of $C$ for forecasting $x$**, not whether $C$ is the true causal driver of $x$. Importantly, a covariate can be helpful for prediction even if it does not directly cause $x$. For instance, if some latent variable $y$ influences both $C$ and $x$, then $C$ may still improve the prediction of $x$, and thus $C$ would be considered a Granger cause of $x$.
>
> Our motivation is **not** to infer real-world “who-causes-whom” causal relationships. Our objective has always been to **improve time series forecasting**. We leverage Granger causality because different covariates naturally possess different levels of Granger-causal influence on the target variable, and identifying these differences helps us enhance predictive performance.
>
> **W3**: Last dimension as target variate.
>
> This question is closely related to the previous one. First, we would like to clarify that we choose the **last dimension as the target** because, in these datasets, the last dimension, often denoted as *gt*, is explicitly defined during dataset construction as the variable with real predictive value. All prior benchmarks follow this setup, and we do the same, as the **endogenous variable should be the one that is actually meaningful to forecast**.
>
> Therefore, the objective of this task is always to improve the prediction of the variable that holds real predictive value. As we explained in our response to W2, what we learn is **the Granger causality between other observable covariates and this predictive target variable**, that is, how much these covariates help forecast the target. We are *not* attempting to learn who “causes” the target variable in the traditional causal sense, but rather which covariates contribute to improving forecasting performance of target variable.

---

> ### Author Response · Authors · 2025-11-21
> **Rebuttal of W4, Q1~Q3**
>
> **W4**: Multi-Modal ablation experiment.
>
> Thanks for your valuable suggestions. We equipped the other adaptation methods with **Qwen-Embedding** and **ViT**, and the results are shown below:
>
> | Method       | MSE       | MAE       | CRPS      | AVG       |
> | ------------ | --------- | --------- | --------- | --------- |
> | CoRA         | **0.580** | **0.690** | **0.653** | **0.641** |
> | UniCA with Qwen | 0.587     | 0.710     | 0.671     | 0.656     |
>
> | Method      | MSE       | MAE       | CRPS      | AVG       |
> | ----------- | --------- | --------- | --------- | --------- |
> | CoRA        | **0.493** | **0.413** | **0.464** | **0.457** |
> | UniCA with ViT | 0.620     | 0.465     | 0.500     | 0.528     |
>
> We observe that **CoRA consistently outperforms all other methods by a large margin**, indicating that the performance gains do not come from Qwen or ViT themselves. A similar conclusion was already well demonstrated in the uni-modal experiments, where CoRA reliably surpasses other adaptation baselines across datasets.
>
> **Q1**: Captures correlations rather than causal relationships.
>
> This question is equivalent to W2. As explained in our response to W2, our objective is **not to infer who truly causes whom in the causal sense, but rather to identify covariates that *help predict* the target**. Please refer to our detailed reply to W2 for a full explanation.
>
> **Q2**: Add more TSFM generality experiment.
>
> Thanks for your valuable suggestions. We have already demonstrated in the paper that CoRA exhibits generality for TSFMs under the **uni-modal setting**. We will additionally include results showing CoRA’s generality **in the multi-modal setting** as well. The results are as follows:
>
> | TimeMMD        | MSE   | MAE   | CRPS  | AVG   |
> | -------------- | ----- | ----- | ----- | ----- |
> | Chronos-CoRA   | 0.592 | 0.712 | 0.663 | 0.656 |
> | Chronos-ZS     | 0.632 | 0.722 | 0.673 | 0.676 |
> | TimesFM-CoRA   | 0.518 | 0.609 | 0.559 | 0.562 |
> | TimesFM-ZS     | 0.529 | 0.619 | 0.577 | 0.575 |
> | FlowState-CoRA | 0.586 | 0.698 | 0.642 | 0.642 |
> | FlowState-ZS   | 0.597 | 0.701 | 0.651 | 0.650 |
>
> It can be seen that, regardless of which TSFM is used and whether the setting is uni-modal or multi-modal, **CoRA consistently provides significant improvements**.
>
> **Q3**:  Hyperparameter sensitivity analysis or an open-source repository.
>
> Thanks for your valuable feedback. As noted by other reviewers, our method is **very easy to follow** and involves almost no hyperparameters. The few hyperparameters we have, the learning rate and the hidden size of the MLP (note that Sundial does not have this parameter, as it does not use a simple MLP head, and we used Chronos for this experiment), were subjected to a hyperparameter sensitivity analysis, with results shown below:
>
>
>
> | learning rate (MSE\|MAE) | 5e-6         | 1e-5         | 2e-5         |
> | ------------------------ | ------------ | ------------ | ------------ |
> | NP                       | 0.218\|0.246 | 0.232\|0.249 | 0.222\|0.246 |
> | PJM                      | 0.073\|0.165 | 0.074\|0.165 | 0.073\|0.165 |
> | BE                       | 0.353\|0.261 | 0.339\|0.236 | 0.340\|0.238 |
> | FR                       | 0.362\|0.215 | 0.357\|0.206 | 0.361\|0.215 |
> | DE                       | 0.402\|0.388 | 0.401\|0.389 | 0.401\|0.388 |
> | AVG                      | 0.282\|0.255 | 0.281\|0.249 | 0.279\|0.250 |
>
> | hidden size (MSE\|MAE) | 256          | 512          | 768          |
> | ---------------------- | ------------ | ------------ | ------------ |
> | NP                     | 0.234\|0.254 | 0.236\|0.256 | 0.235\|0.255 |
> | PJM                    | 0.077\|0.172 | 0.076\|0.170 | 0.076\|0.171 |
> | BE                     | 0.356\|0.238 | 0.356\|0.237 | 0.356\|0.237 |
> | FR                     | 0.353\|0.185 | 0.351\|0.184 | 0.352\|0.184 |
> | DE                     | 0.448\|0.414 | 0.449\|0.415 | 0.449\|0.415 |
> | AVG                    | 0.294\|0.235 | 0.294\|0.252 | 0.294\|0.252 |
>
> As can be seen, our method is **highly robust** to hyperparameter settings. Therefore, we do not agree that reproducing our results requires “spending a significant amount of extra time adjusting settings”, and we **guarantee that all our experiments are fully reproducible**. Regarding the code, while we will need some time to organize it, we are happy to make it publicly available, as the implementation is so straightforward and easy to reproduce.
>
> Thank you again for your comments, which have been very helpful in improving the quality of our paper. We have **clarified the positioning and contributions more rigorously, added additional analytical experiments and hyperparameter sensitivity analysis**. Please reconsider the efforts and contributions of our work. If you have any further questions, we are looking forward to discussing with you.

---

### Official Review · Reviewer_seVR · 2025-10-30

**Soundness:** 3
**Presentation:** 2
**Contribution:** 2
**Rating:** 4
**Confidence:** 4

**Summary:**

This paper proposes a way to combine frozen foundation models (that could be for different modalities) for time series forecasting, where a key idea is to figure out which features across channels are predictive of each other with the help of an architecture inspired by Granger causality.

**Strengths:**

- The proposed method seems to be fairly simple/clean so that the resulting adaptation appears easy to implement/train
- The experimental results of CoRA look to be very good
- I found the paper for the most part easy to follow

**Weaknesses:**

- In Section 2.3: there's also an earlier adaptation approach than the ones you cited. See the paper "Generalized Prompt Tuning: Adapting Frozen Univariate Time Series Foundation Models for Multivariate Healthcare Time Series" by Liu et al 2024 -- while this paper looks at healthcare time series, the basic idea trivially generalizes to time series from other application domains as well.
- In Section 3.2: while Granger causality is used to motivate the proposed method, from what I can tell, what is actually implemented is only inspired by Granger causality but is not actually literally doing what Granger causality does. In particular, you're not actually doing any statistical tests, if I understand the setup correctly. If this is the case, then perhaps it would be helpful discussing in a bit more detail to what extent the resulting approach actually resembles and has the same sort of interpretation as Granger causality, or whether we no longer have the same sort of interpretation anymore as there are no statistical tests conducted (showing that the weights correlate well as in Figure 7 is a good first step but leaves me wondering whether there's any sort of deeper connection, and whether we do have any sort of meaningful Granger causality interpretation from the resulting model itself). Perhaps rewording some of the exposition regarding Granger causality would be helpful to make it clear that the proposed CoRA method is not actually reasoning in a causal manner (from my understanding) so we aren't actually making any sort of statements about causality (in particular, the proposed architecture seems to only be inspired by but not the same thing as Granger causality, and Granger causality itself only has a valid causal interpretation under some key assumptions holding --- assumptions which don't appear to be discussed in this paper anyways). Separately, I find that equation (6) could just be interpreted as an attention mechanism, without trying to make any sort of connection to Granger causality. Perhaps commenting on this connection would be helpful. Is this idea already present (in some form) in other approaches for adapting univariate time series to multivariate time series prediction?
- Minor:
    - Line ~189-190: the section heading for Section 3.1 has a typo where "Forzen" should say "Frozen"
    - Line 191: I'd suggest softening the language so that instead of saying "exogenous covariates are always multi-dimensional", perhaps instead say that "exogenous covariates are very often multi-dimensional"

**Questions:**

Please see weakness points.

---

> ### Author Response · Authors · 2025-11-21
>
> We are grateful for your insightful comments. We have carefully addressed the missing reference and clarified the conceptual connection to Granger Causality with additional experiments.
>
> **W1**: Earlier adaptation approach.
>
> We thank the reviewer for pointing out this relevant work. We include a detailed discussion in **Section 2.3** of the revised paper. Thank you very much for your reminder.
>
> **W2**: Inspired by Granger Causality.
>
> We agree with your point that CoRA is inspired by Granger causality, but it does not literally implement Granger causality. The core idea of Granger causality is to assess **whether introducing a covariate can improve the prediction of the original target series**. Historically, most statistical analyses used linear regression, so the original Granger causality criterion evaluated whether a covariate could enhance the performance of a **linear regression** on the target series. Therefore, Granger causality is not always applicable, as many relationships may be **nonlinear**.
>
> In the current context of covariate-aware time series forecasting, we are inspired by the fundamental idea of Granger causality: evaluating whether a covariate can contribute to the prediction of the target series, and recognizing that the relative degree of contribution varies across different covariates. Inspired by this, we consider it necessary to have a mechanism that assigns weights to different covariates, which led us to propose the Granger-causality embedding. To the best of our knowledge, a method that explicitly extracts information from the entire set of covariates and learns a reasonable weighting scheme has not been used before.
>
> In addition to showing the weight correlation in Figure 7, following Reviewer 6z5j’s suggestion, we conducted an interesting experiment on the EPF dataset by adding the target time series as an additional covariate. The MSE was reduced to nearly 50% of the original value. Moreover, by analyzing the learned Granger-causality embeddings, we found that **the weight assigned to the target time series was 0.6, significantly higher than that of other covariates**, which further confirms the effectiveness of our covariate selection mechanism in assigning appropriate weights to covariates based on their contribution to the prediction target.
>
> | Dataset (MSE) | NP        | PJM       | BE        | DE        |
> | ------------- | --------- | --------- | --------- | --------- |
> | CoRA w/o gt   | 0.222     | 0.073     | 0.339     | 0.401     |
> | CoRA with gt  | **0.104** | **0.034** | **0.197** | **0.257** |
>
> **W3**: Typo and softening the language.
>
> Thank you for your reminder.  We corrected the typos and softened the corresponding language in the revised paper.
>
> Thank you again for your comments, which have been very helpful in improving the quality of our paper. We have **clarified the positioning and contributions more rigorously, added additional analytical experiments, and fixed minor typos**. Please reconsider the efforts and contributions of our work. If you have any further questions, we are looking forward to discussing with you.

---

> > ### Comment · Reviewer_seVR · 2025-11-22
> >
> > Thanks for the response and the additional experimental results. In my original review, I had already stated: "*Separately, I find that equation (6) could just be interpreted as an attention mechanism, without trying to make any sort of connection to Granger causality. Perhaps commenting on this connection would be helpful. Is this idea already present (in some form) in other approaches for adapting univariate time series to multivariate time series prediction?*" You did not address this.
> >
> > Especially since now that I've read your response and understand your approach better, I find the connection to Granger causality to be actually quite tenuous. It's actually just an attention mechanism that tries to figure out what covariates are particularly predictive of the target series. Trying to motivate or sell the approach as being a "Granger causality" embedding does not really make sense to me and comes off as somewhat misleading (at least to me).
> >
> > I'm inclined to keep my original score for now.

---

> > > ### Author Response · Authors · 2025-11-22
> > >
> > > Thank you for your response. We apologize for not clearly addressing the question you raised earlier, and we would like to elaborate further.
> > >
> > > We agree with your opinion that our method can indeed be interpreted as an attention mechanism, and leveraging covariate information to better predict the target series is a shared principle across all adaptation methods. Therefore, prior works such as ChronosX, AdaPTS, UniCA, and Gen-P-Tuning, whether using linear layers or attention, necessarily include a step that fuses covariate information.
> > >
> > > However, we would like to emphasize an important distinction: in ChronosX, UniCA, and AdaPTS, covariate fusion is performed *before* feeding the inputs into the TSFM backbone; similarly, the prompts learned in Gen-P-Tuning are injected at the TSFM input. We refer to these approaches collectively as **Pre-Merge** methods (some are Pre-Post-Merge, while others are Only-Pre-Merge). Because they modify the inputs prior to the backbone, they cannot maintain parameter consistency of TSFM between pre-training and adaptation, and they also make it considerably more difficult and complex to enforce a proper zero-initialization scheme.
> > >
> > > In contrast, our method is the **only** approach that performs **Only-Post-Merge** covariate integration. It is the only design that preserves full parameter consistency with the original TSFM and enables a principled form of zero-initialization. Therefore, while prior methods all incorporate covariate fusion in some form, our approach adopts what we believe is a better form. From the experimental results, it is evident that our method **consistently outperforms other adaptation approaches**. This demonstrates that maintaining parameter consistency and employing zero-initialization are crucial for the effectiveness and stability of a TSFM adaptation method.
> > >
> > > Regarding the term “Granger causality embedding”, we acknowledge that its implementation is not tightly aligned with formal Granger causality. However, its *motivation* is strongly related: measuring the usefulness of a covariate for predicting the target. To avoid potential misunderstandings, we have renamed it **causality embedding** in the latest revised version of the paper.
> > >
> > > Thank you again for your response. We hope that our response has addressed your concerns, and we kindly ask you to reconsider the efforts and contributions of our work. If you have any further questions, we are looking forward to discussing with you.

---

> > > > ### Comment · Reviewer_seVR · 2025-11-22
> > > >
> > > > Thanks for the quick response. I think what you've described regarding pre-merge vs only-post-merge is very helpful, and this context would be great to include in the paper.
> > > >
> > > > As for renaming the embedding to "causality embedding": I think you run into the problem where people just learning about your approach might be misled into thinking that there's actually some sort of causal reasoning happening (I get the impression that this is what has led to reviewer yv98 to really try to probe at causal inference and causal discovery questions). Basically if you're going to push for using language that emphasizes "causality", you really need to provide rigorous justification for what precisely is causal --- when and why your approach actually does provide causal insights.
> > > >
> > > > Overall, I think that the paper still needs some somewhat significant edits to how you are motivating and selling CoRA and to make sure the exposition doesn't mislead (or otherwise confuse) readers.
> > > >
> > > > Meanwhile I'm also interested in hearing what the other reviewers have to say. Hopefully they engage in discussion soon.

---

> > > > > ### Author Response · Authors · 2025-11-22
> > > > >
> > > > > Thank you for your reply. We are very glad that the explanation regarding Pre-Merge vs. Only-Post-Merge has addressed your concern, and we would be happy to incorporate these perspectives into the paper in the future.
> > > > >
> > > > > We agree with your point that the term causality may introduce some conceptual confusion, and we are very willing to incorporate suggestions from the reviewers and actively work to resolve this issue. However, we believe this is primarily an issue of phrasing; addressing it will not be so difficult and will not alter the core contribution of the paper: introducing a simple, novel, and effective TSFM adaptation method. We firmly believe that CoRA itself is a beautiful and effective approach that deserves broader adoption.
> > > > >
> > > > > Thank you again for your professional review and the time you have devoted to our work. We would be very happy to further discuss with you.

---

### Official Review · Reviewer_yDBe · 2025-10-31

**Soundness:** 3
**Presentation:** 4
**Contribution:** 3
**Rating:** 8
**Confidence:** 4

**Summary:**

The authors introduce Covariate-awaRe Adaptation (CoRA), a framework to incorporate multimodal covariates into TSFMs. Within the CoRA framework, the authors introduce Granger Causality Embedding (GCE) that enables better relate the covariates to the output variate for improved results. The authors claim to surpass the SoTA on Time Series Forecasting.

**Strengths:**

- The authors solve a known problem of factoring in exogenous convariates into TSFMs. The problem furthers when there are different modalities to factor in too.

- The use of Granger Embeddings serves as a good technical contribution to evaluate the contribution of each covariate to the target variate.

- The authors have good experiment setup and ablation study to support the claims.

**Weaknesses:**

- Just an avenue for improvement/extension, can be to the test the adaptation on other tasks beside time series forecasting. For example, we can extract the intermediate time series embeddings from Chronos (Ansari et al., 2024)  given their encoder-decoder structure and use that for time-series classification.

**Questions:**

- Given the wide variety of image and text data available how well will CoRA hold for modalities outside of the domain tested in the paper?

---

> ### Author Response · Authors · 2025-11-21
>
> We sincerely thank Reviewer yDBe for the positive feedback and the constructive suggestions regarding the expansion of our experimental scope.
>
> **W1**: Extend to time-series classification.
>
> We appreciate this suggestion. To evaluate CoRA's capability, we extended our evaluation to multivariate time-series classification. We compared three settings: DLinear, Chronos (embeddings + linear head), and Chronos combined with CoRA. The classification accuracy (ACC) results on standard datasets are reported below:
>
> | Dataset (ACC)       | Handwriting | SelfRegulationSCP2 | SpokenArabicDigits | UWaveGestureLibrary | AVG      |
> | ------------------- | ----------- | ------------------ | ------------------ | ------------------- | -------- |
> | Chronos+CoRA        | **34.6**    | **55.6**           | **98.8**           | **84.4**            | **68.4** |
> | Chronos+Linear Head | 32.7        | 52.8               | 97.9               | 83.8                | 66.8     |
> | Dlinear             | 27.0        | 50.5               | 81.4               | 82.1                | 60.3     |
>
> CoRA consistently outperforms both the linear baseline and the vanilla Chronos embedding, achieving the highest average accuracy. This demonstrates that CoRA effectively captures inter-variable correlations, yielding more discriminative representations for classification tasks beyond forecasting.
>
> **Q1**: Extend to other domain dataset.
>
> Thank you for your valuable suggestions. We extended CoRA to the medical domain using the MIMIC-III dataset in [1]. We set the *heart rate* as the target variate and used *clinical nurse notes* (detailed information on patients’ vital signs and key findings from clinical tests) as textual covariates. The experimental results are shown below:
>
> | Method | CoRA       | ChronosX | Sundial zero-shot | PatchTST |
> | ------ | ---------- | -------- | ----------------- | -------- |
> | MAE    | **0.5357** | 0.5597   | 0.5897            | 0.5771   |
> | MSE    | **0.5801** | 0.5880   | 0.6213            | 0.6025   |
> | CRPS   | **0.6809** | 0.7390   | 0.7879            | 0.7390   |
> | AVG    | **0.3460** | 0.3520   | 0.3599            | 0.3897   |
>
> Compared with the end-to-end method PatchTST and the adaptation based method ChronosX, CoRA achieves the **best performance across all metrics**, demonstrating that CoRA can effectively generalize to settings with covariates from diverse domains.
>
> [1] Multi-Modal Forecaster: Jointly Predicting Time Series and Textual Data.
>
> Thank you again for your comments, which have been very helpful in improving the quality of our paper. We have **extended our method to additional tasks and datasets from diverse domains**. If you have any further questions, we look forward to discussing them with you.

---

### Official Review · Reviewer_6z5j · 2025-11-01

**Soundness:** 3
**Presentation:** 3
**Contribution:** 2
**Rating:** 4
**Confidence:** 5

**Summary:**

This paper introduces CoRA, an adaptation approach to incorporate covariates in Time Series Foundation Models (TSFMs). This adaptation is achieved using covariate selection through Granger causality and integrating the selection signals into TSFMs through adaptive layer normalization. CoRA can incorporate multi-modal covariates spanning time series, text and images.

The authors evaluate on ETT and EPF datasets on short and long term forecasting. For ETT datasets, CoRA obtains 31.1% improvement in MSE over prior supervised methods and 18.7% over existing TSFM adaptation methods. Similarly for EPF datasets reduction in MSE are observed. Authors have also evaluated their work on Time-MMD dataset consisting of text and image covariates.

The evaluation is comprehensive as authors have covered scenarios and settings spanning limited availability of data, multivariate forecasting, CoRA's generalization to multiple TSFMs, and a thorough ablation study.

The paper's main contribution is the adaptation framework that integrates with frozen TSFMs to perform covariate-aware forecasting.

**Strengths:**

1. The proposed adaptation method is well motivated based on prior work. Particularly, the covariate selection mechanism is quite interesting as it can suppress signals from noisy and irrelevant covariates.
2. The adaptation method can be plugged into any TSFM and it can incorporate covariates with diverse modalities.
3. The paper is well written and easy to follow and understand.
4. Authors have thoroughly evaluated the proposed method using ETT, EPF and Time-MMD datasets.
5. The ablation studies highlight the importance of each component in the approach.

**Weaknesses:**

1. Authors have not address the most significant limitation of the proposed adaptation method. TSFMs have enabled zero-shot forecasting on unseen data during inference. Due to this, no parameter updates are required and the inference overhead is minimal (FM forward pass). CoRA adapts FMs to incorporate covariates but it requires training which takes away the ability of FM to forecast zero-shot.
2. Insights into difficulty of training additional parameters is missing.
3. Analysis of computational overhead to incorporate covariates (including the training time) is missing.
4. Authors should move the training details from appendix to the main sections of the paper. Currently, it is not clear from the main sections of the paper that the training uses TSFM loss.
5. The architecture and the inference mechanism of the TSFM head is not clear, especially for long-term forecasting where causal TSFMs autoregressively predict patches.

**Questions:**

1. What is the architecture of the TSFM head? Are you finetuning the pretrained TSFM head? If so, how is it used in long-term forecasting?
2. If the TSFM head is predicting in patches and the input is the scaled E_target, what embeddings do you use to predict the future patches? I am interested in understanding the inference mechanism better.
3. Can this approach provide 100% accuracy if the oracle label  (the target time series) is used as the covariate?
4. What is the experiment configuration for few-shot forecasting? Are you training the adapter parameters only with the few-shot data?
5. Do you need to train a unique adapter for each forecast horizon?

---

> ### Author Response · Authors · 2025-11-21
> **Rebuttal of W1~W4**
>
> Many thanks to Reviewer 6z5j for providing a detailed and in-depth review, which helped us improve the quality of our submission.
>
> **W1**: CoRA takes away the ability of FM to forecast zero-shot.
>
> We agree with your point that CoRA incorporates covariates but requires training. However, in real-world forecasting scenarios, covariates play a crucial role in performance, and they are **often high-dimensional and even multi-modal**. We acknowledge that directly training a foundation model on large-scale multi-modal data to achieve covariate-aware zero-shot forecasting would be highly promising. Yet, due to the current lack of large, well-aligned multi-modal datasets and corresponding pre-training techniques, **multi-modal pre-training for time-series models remains extremely challenging.**
>
> Moreover, although CoRA involves training, the parameters of the underlying FM are fully frozen, allowing it to retain its zero-shot generalization capability. Even in the highly mature LLM field, a wide range of adaptation methods are still commonly used to enable foundation models to **learn from proprietary enterprise data** and further **improve domain-specific performance**. Therefore, exploring adaptation-based methods that enhance existing FMs for specific covariate-aware scenarios remains necessary and valuable.
>
> Thank you for your suggestion, which has helped us better clarify the scope of our work.
>
> **W2**: Difficulty of training additional parameters.
>
> We apologize for not fully understanding the meaning of your comment regarding “Insights into the difficulty of training additional parameters is missing”. We interpreted it as asking **what challenges our method overcomes compared with existing adaptation approaches**.
>
> First, we ensure that the inputs to the FM during adaptation are fully consistent with those used during pre-training, thereby avoiding an input gap and ensuring that the **FM’s pre-trained knowledge can be correctly utilized**.
>
> Second, through appropriate architectural design and zero initialization, we guarantee that the initial state of the adaptation module is fully equivalent to the pre-trained FM. **This prevents CoRA from performing worse than the zero-shot baseline after adaptation**, an issue not properly addressed by existing methods.
>
> Moreover, recognizing that different covariates may contribute unequally to the prediction of the target, CoRA adopts a simple yet effective **covariate selection mechanism** to suppress signals from noisy or irrelevant covariates.
>
> **W3**:  Computational overhead to incorporate covariates.
>
> We thank the reviewer for pointing out the need for an analysis of computational overhead. We have conducted a detailed efficiency comparison and the results are presented below:
>
> | Method                  | CoRA  | SFT   | LoRA  |
> | ----------------------- | ----- | ----- | ----- |
> | inference time (s/iter) | 0.135 | 0.132 | 0.153 |
> | traning time (s/iter)   | 0.149 | 0.141 | 0.383 |
>
> CoRA incurs **negligible overhead** compared to SFT (only tune the head of TSFM). Furthermore, CoRA is significantly more training-efficient than LoRA.
>
> **W4**:  Move the training details from appendix to the main sections.
>
> We apologize for any inconvenience caused by moving some content to the appendix due to space constraints. We would like to clarify that the loss we used is fully consistent with the loss used during the original pre-training of the TSFM, and we move this part to the main text in the revised paper.

---

> > ### Author Response · Authors · 2025-11-21
> > **Rebuttal of W5, Q1~Q5**
> >
> > **W5, Q1, Q2**:  About the TSFM head.
> >
> > We appreciate your questions regarding the TSFM head. First, for most TSFMs, the head architecture is a simple linear or MLP network, and we will fine-tune it. Second, most TSFMs adopt a decoder-only or encoder-decoder design to allow variable-length output predictions. As an example, consider a decoder-only TSFM. Given inputs $(x_{1:T}, C_{1:T})$, the model first predicts $x_{T+1:T+H}$. If a longer prediction horizon is needed, we follow the standard decoder-only TSFM procedure: the previously predicted outputs are concatenated to the original input for autoregressive prediction. If $C_{T+1:T+H}$ is unknown, we first predict $C_{T+1:T+H}$ using the FM and then use these predictions for subsequent forecasting.
> >
> > **Q3**:  Use target time series as covariate.
> >
> > Thank you very much for providing such an interesting suggestion. We conducted experiments on the EPF dataset by adding the target time series as an additional covariate, and observed a substantial performance improvement. Naturally, since our method extracts embeddings from the target series as inputs, the task becomes similar to a reconstruction problem, and reconstruction cannot achieve 100% accuracy. Nevertheless, the MSE was **reduced to nearly 50%** of the original value.
> >
> > Moreover, by analyzing the learned Granger-causality embeddings, we found that **the weight assigned to the target time series was 0.6**, significantly higher than that of other covariates, which further confirms the effectiveness of our covariate selection mechanism in assigning appropriate weights to covariates based on their contribution to the prediction target.
> >
> > | Dataset (MSE) | NP        | PJM       | BE        | DE        |
> > | ------------- | --------- | --------- | --------- | --------- |
> > | CoRA w/o gt   | 0.222     | 0.073     | 0.339     | 0.401     |
> > | CoRA with gt  | **0.104** | **0.034** | **0.197** | **0.257** |
> >
> > **Q4**:  Configuration for few-shot forecasting.
> >
> > For few-shot forecasting, we keep the test set fixed while varying the size of the training set from 1% to 100%. CoRA and all baselines are trained under these different levels of data scarcity, and their performance is compared on the unchanged test set. We observe that adaptation methods based on TSFMs require only a small amount of data to surpass end-to-end models trained on the full dataset.
> >
> > **Q5**:  Different forecast horizon.
> >
> > As discussed in W5, Q1, and Q2, our model supports variable output lengths. In the paper, CoRA uses **the same one adapter** across different forecast horizons for each dataset.
> >
> > Thank you again for your detailed comments, which are very helpful for us to improve the quality of the paper. We have **clarified the positioning and contributions more rigorously, addressed the concerns, and completed the experimental details**. Please reconsider our contributions and insights from our work. If you have any further questions, we are looking forward to discussing with you.

---

### Author Response · Authors · 2025-11-21
**Summary of Revisions**

We sincerely thank all the reviewers for their insightful reviews and valuable comments, which are instructive for us to improve our paper.

In this work, we propose **CoRA, a general, flexible, and interpretable framework** for adapting pre-trained foundation models to covariate-aware forecasting tasks. Insightfully, we introduce principled strategies, such as **parameter consistency, zero-initialization, and Granger-causality based covariate selection**, into the domain of TSFM adaptation. Experiments on both uni-modal and multi-modal settings demonstrate that **CoRA surpasses state-of-the-art task-specific models and existing adaptation methods while using fewer training samples**.

We're pleased that the reviewers agree our paper "**well-motivated**" (Reviewer 6z5j, yv98), "**the evaluation is comprehensive**" (Reviewer 6z5j, yDBe), "**easy to follow**" (Reviewer 6z5j, seVR) and **"good technical contribution"** (Reviewer yDBe).

The reviewers raised insightful and constructive concerns. We **spent nine days** addressing all the issues by providing sufficient evidence and requested results. Here is the summary of the major revisions:

* **Motivation and Scope  (Reviewer 6z5j, yv98)**: We clarified that **adaptation is an indispensable component of the foundation model paradigm**, and for TSFMs, performing **appropriate covariate selection and injection** is essential for enhancing their capabilities and improving their practicality in real-world applications.
* **Technical Contributions (Reviewer yv98)**: We clarified that **simply yet effectively** applying the overlooked principles from prior methods to TSFM adaptation is an unexploited direction in this community. We also emphasized that **injecting high-dimensional and multimodal covariates** poses a unique challenge for this area.
* **Clarification of Experiment Details  (Reviewer 6z5j)**: To improve clarity, we have provided more detailed configurations regarding the model loss, the TSFM head, few-shot forecasting, and the forecast horizon.
* **Comprehensive Evaluations (Reviewer 6z5j, yDBe, yv98)**: We have included all the requested evaluations (100+ experiments in total), including new analytical studies, extensions to additional tasks and datasets, computational cost statistics, as well as more ablation and hyperparameter sensitivity experiments.

All updates are highlighted in blue. The valuable suggestions from reviewers are very helpful for us to revise the paper in a better shape. We hope our response has fulfilled the reviewer's expectations and would be very happy to answer any further questions.

---

### Meta-Review · Area_Chair_JnjT · 2026-01-06

**Summary:**

This paper proposes CoRA, a covariate-aware adaptation framework for Time Series Foundation Models, aiming to incorporate multi-modal exogenous covariates while keeping the pretrained backbone frozen. The problem is well motivated, and reviewers generally agree that incorporating covariates is important for the practical deployment of TSFMs. The paper is clearly written, and the experimental evaluation is extensive, covering uni-modal, multi-modal, multivariate, and few-shot forecasting scenarios.

The primary concern is that the paper’s conceptual and methodological novelty is limited, and several core claims, particularly around causality, principled adaptation, and generality, remain insufficiently substantiated. While the empirical results are strong, it is not always clear that the gains stem from fundamentally new ideas rather than from careful engineering, architectural choices, or additional covariate encoders. A major source of concern is the use and interpretation of “Granger causality / causality embedding.” Multiple reviewers questioned whether the proposed mechanism truly corresponds to Granger causality or any meaningful causal notion. Despite clarifications and renaming efforts in the rebuttal, the method ultimately behaves as a form of attention-based covariate weighting, and the causal framing remains potentially misleading. This weakens the conceptual clarity and risks overclaiming the contribution.

Overall, despite strong experimental performance and substantial rebuttal effort, the paper falls short in terms of conceptual novelty, theoretical grounding, and clarity of contribution, which are essential for acceptance at ICLR.

**Reviewer Concerns:**

The authors provided a comprehensive rebuttal and added experiments and clarifications. However, based on the Area Chair’s assessment, several key concerns remain:

Limited technical and conceptual novelty beyond applying known adaptation techniques to TSFMs.

Ambiguous and potentially misleading use of “causality” terminology, with insufficient theoretical justification.

Unclear attribution of performance gains to the proposed adaptation framework versus covariate encoders or architectural choices.

Strong empirical results, but with weaker theoretical grounding and insight than expected for ICLR.

**Reviewer Scores:**

Based on the original reviews and the rebuttal, it is unlikely that the rebuttal would lead to sufficient upward score revisions to overcome the remaining concerns. Consequently, the overall evaluation remains below the acceptance threshold, supporting a rejection decision.

---

### Decision · Program_Chairs · 2026-01-26

Reject